# Major vault protein suppresses obesity and atherosclerosis through inhibiting IKK–NF-κB signaling mediated inflammation

Jingjing Ben[1], Bin Jiang[1], Dongdong Wang[1], Qingling Liu[1], Yongjing Zhang[1], Yu Qi[1], Xing Tong[1], Lili Chen[1], Xianzhong Liu[2], Yan Zhang[1], Xudong Zhu[1], Xiaoyu Li[1], Hanwen Zhang[1], Hui Bai[1], Qing Yang[1], Junqing Ma[1], Erik A.C. Wiemer [ID] [3], Yong Xu[1] & Qi Chen[1]

Macrophage-orchestrated, low-grade chronic inflammation plays a pivotal role in obesity and atherogenesis. However, the underlying regulatory mechanisms remain incompletely understood. Here, we identify major vault protein (MVP), the main component of unique cellular ribonucleoprotein particles, as a suppressor for NF-κB signaling in macrophages. Both global and myeloid-specific *MVP* gene knockout aggravates high-fat diet induced obesity, insulin resistance, hepatic steatosis and atherosclerosis in mice. The exacerbated metabolic disorders caused by MVP deficiency are accompanied with increased macrophage infiltration and heightened inflammatory responses in the microenvironments. In vitro studies reveal that MVP interacts with TRAF6 preventing its recruitment to IRAK1 and subsequent oligomerization and ubiquitination. Overexpression of MVP and its α-helical domain inhibits the activity of TRAF6 and suppresses macrophage inflammation. Our results demonstrate that macrophage MVP constitutes a key constraint of NF-κB signaling thereby suppressing metabolic diseases.

[1] Department of Pathophysiology, Key Laboratory of Cardiovascular Disease and Molecular Intervention, Nanjing Medical University, Nanjing 211166, China. [2] Department of General Surgery, Bayi Clinical Medicine School, Nanjing Medical University, Nanjing 210002, China. [3] Department of Medical Oncology, Erasmus MC Cancer Institute, Erasmus University Medical Center, Rotterdam 3000 CA, The Netherlands. These authors contributed equally: Jingjing Ben, Bin Jiang, Dongdong Wang. Correspondence and requests for materials should be addressed to J.B. (email: bjj@njmu.edu.cn) or to Q.C. (email: qichen@njmu.edu.cn)

Low-grade, chronic inflammation is implicated in many immune-metabolic diseases including obesity and athero-sclerosis[1–4]. The macrophage is an important immune cell orchestrating chronic inflammatory responses by sensing and reacting to various stresses in metabolic organs including adipose tissue, liver, and artery wall[4–9]. Inflammatory cytokines and chemokines, such as tumor necrosis factor (TNF)-α, interleukin (IL)-6, IL-1β, and C–C motif ligand-2 (CCL2), disrupt metabolic homeostasis and the functions of metabolic cells and stromal components[1,2,4,5]. Thus, inflammatory responses determine the metabolic pathophysiological outcome in the diseased microenvironment.

Inflammatory signaling in cells is composed of receptors, signaling kinases, and effectors. Pattern recognition receptors (e.g., toll-like receptors, TLRs) play pivotal roles in both the initiation and the resolution of inflammation[10,11]. For example, the activation of TLR4 can stimulate the myeloid differentiation primary response gene 88 (MyD88)-dependent signaling and promotes the assembly of a complex containing the interleukin 1 receptor-associated kinase 1 (IRAK1) and the TNF receptor-associated factor 6 (TRAF6), which results in the activation of IκB kinases (IKKs) and, eventually, transcription factor nuclear factor κB (NF-κB)[11–14]. IKK–NF-κB cascades have been implicated in immune-mediated and inflammatory diseases[15–18]. As the immune system needs to constantly strike a balance between activation and inhibition to avoid detrimental and inappropriate inflammatory responses, pro-inflammatory signaling like the NF-κB pathway must be tightly regulated. Although the mechanisms of NF-κB activation have been well studied, the intrinsic negative regulatory mechanisms in the inflammatory response need to be further explored.

Major vault protein (MVP) is the main component of cellular ribonucleoprotein particles known as vaults[19]. The unique vault structure, consisting of 78 MVP subunits, numerous copies of the vault-associated proteins including vault poly(ADP-ribose) polymerase (VPARP) and telomerase-associated protein-1 (TEP1), and small untranslated RNA (vRNA), is implicated in the regulation of several cellular processes including nucleocyto-plasmic transport, signaling transduction, cellular differentiation, cell survival, and immune responses[19–25]. In the present study, we investigate the role of MVP in metabolic inflammation. By using several animal models of metabolic diseases, we identify macrophage MVP as an important suppressor of NF-κB activation by preventing TRAF6 ubiquitination. This consequently inhibits NF-κB pathway-related metabolic inflammation and attenuates obesity-associated insulin resistance, hepatic steatosis, and atherosclerosis. The discovery of MVP-mediated negative regulation of NF-κB may pave the way for clinical intervention strategies for metabolic diseases.

## Results

**Macrophage MVP is up-regulated in obese adipose tissues.** Obesity is a central feature of metabolic diseases. To understand the role of MVP in metabolic diseases, we firstly determined the role of MVP in obesity. Obese male C57BL/6J mice were generated by administering with a high-fat diet (HFD) for 12 weeks. We found that obesity caused a significant increase of MVP in the epididymal white adipose tissue (epiWAT) (Fig. 1a), particularly in the stromal vascular fraction cells (SVFs) but not in the adipocytes of epiWAT (Fig. 1b). This differential expression pattern of MVP was reproduced in the isolated SVFs and adipocytes from epiWAT by western blot analysis. Expression levels of MVP were found to be dramatically higher in the isolated SVFs than in adipocytes in both normal chow diet (CD)- and HFD-fed mice

(Fig. 1c). Immunofluorescence staining revealed that MVP co-localized mainly with CD68[+] macrophages in the adipose tissue (Fig. 1d). When F4/80[+] macrophages were isolated from epiWAT SVFs by flow cytometry, we confirmed the HFD-induced overexpression of MVP in macrophages (Fig. 1e). Consistently, significant increased MVP levels were detected in HFD-fed murine peritoneal macrophages (PMs) compared with CD-fed murine PMs (Fig. 1f). MVP was also up-regulated in gonadal WAT (gonWAT) macrophages and PMs from the HFD-fed female mice (Supplementary Fig. 1a, b), suggesting a similar expressional trend of MVP in both male and female obese mice.

We next measured the expressional level of MVP in obese human beings. Immunohistochemistry (IHC) staining showed that the expression of MVP in the stromal compartment was substantially increased in the visceral adipose tissue of overweight or obese individuals compared with normal weight controls (Fig. 1g, h). There was a co-localization of MVP with CD68[+] macrophages in human visceral adipose tissues (Fig. 1i). MVP expression was much higher in the CD14[+] macrophages isolated from visceral adipose tissue in overweight or obese persons than in normal weight individuals (Fig. 1j).

In summary, up-regulation of MVP expression in macrophages of visceral adipose tissues was correlated with obesity in both humans and mice, suggesting that MVP be involved in obesity-associated inflammation.

**MVP deficiency aggravates obesity and metabolic disorders.** We further deleted the *MVP* gene (*MVP* KO) in male mice which were fed with either a CD or a HFD for 7 weeks together with the age- and sex-matched wild-type (WT) littermates. MVP deficiency did not influence murine body weight and glycolipid metabolism under CD-fed conditions (Supplementary Fig. 2). However, upon HFD challenge, *MVP* KO mice gained more weight (Fig. 2a) and displayed a higher weight of multiple adipose depots (Fig. 2b) including epi, mesenteric (m), perirenal (peri), subcutaneous (sub) WAT, and brown adipose tissue (BAT) than WT mice. Larger adipocyte size and lower expression of *adipo-nectin* (*ADIPOQ*) and *leptin* (*LEP*), the known important adi-pokines produced by functional adipocytes, were detected in HFD-fed *MVP* KO mice (Fig. 2c–e).

Obesity impairs glucose metabolism in the body. In the present study, we found that MVP deficiency exacerbated HFD-induced high blood glucose (Fig. 2f) and glucose-induced hyperinsuline-mia (Fig. 2g). Furthermore, MVP deficiency impaired glucose tolerance and insulin tolerance in mice (Fig. 2h, i) in conjunction with suppressed phosphorylation of AKT, a readout of intracel-lular insulin signaling, in epiWAT, liver, and skeletal muscle (Fig. 2j). These data suggest that MVP deficiency may aggravate obesity-associated insulin resistance in mice.

We also examined the impact of MVP deficiency on lipid metabolism. It was shown that plasma levels of nonesterified fatty acid (NEFA), triglycerides (TG), and total cholesterol (TCH) were significantly increased in HFD-fed *MVP* KO mice compared with WT mice (Fig. 2k, l). Hepatic steatosis is nearly a uniform feature of obesity. Indeed, we found that MVP deficiency caused a dramatic increase in liver weight, intra-hepatic TG and TCH contents, and plasma levels of AST and ALT (Fig. 2m–o). Consistently, a dramatic change in the overall liver morphology with accumulation of large droplet-like structures (Fig. 2p, q) and higher expression of the fatty acid synthesis and uptake genes (*FASN*, *SCD1*, *SREBP-1C*, *PPARγ*, and *CD36*) in the liver (Fig. 2r) were observed in *MVP* KO mice. HFD-fed female *MVP* KO mice exhibited similar phenotypic changes to those male mice (Supplementary

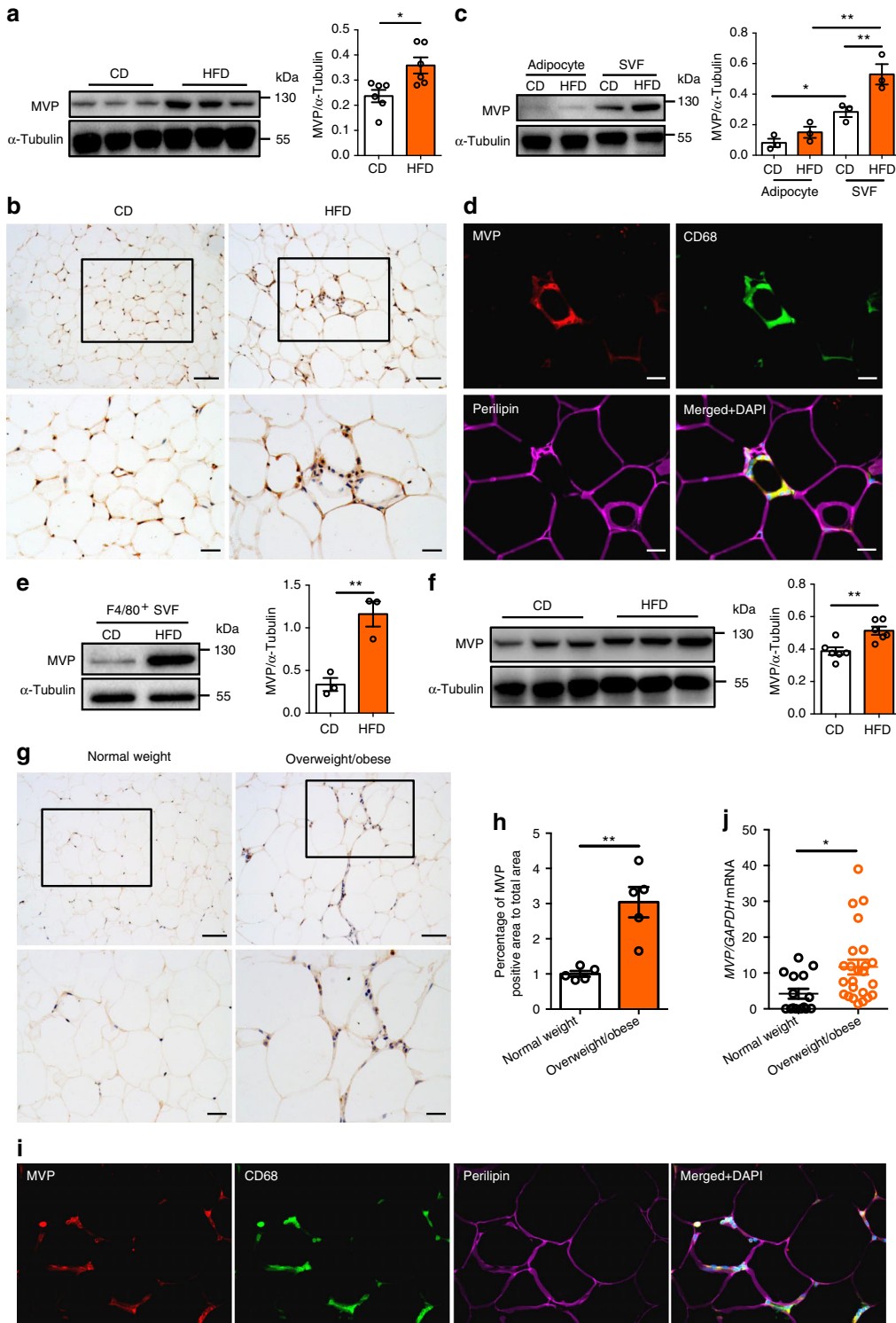

**Fig. 1** MVP expression is up-regulated in macrophages from obese mice and human beings. Male C57BL/6J mice were fed a CD or a HFD for 12 weeks. **a** Western blot analysis of MVP expression in epiWAT (n = 6). **b** IHC staining of MVP in epiWAT. Scale bars, 50 μm (top) and 20 μm (bottom). **c** Western blot analysis of MVP expression in the adipocytes and SVFs isolated from epiWAT (n = 3). **d** Immunofluorescence images of staining with antibodies against CD68 (green), MVP (red), and Perilipin (purple) in epiWAT of HFD-fed mice. Nuclei were stained with DAPI (blue). Scale bars, 20 μm. **e** Western blot analysis of MVP expression in sorted F4/80[+] macrophages isolated from epiWAT SVFs (n = 3). **f** Western blot analysis of MVP in PMs (n = 6). **g**, **h** IHC staining (**g**) and quantitative analysis (**h**) of MVP in the visceral adipose tissue from normal weight donors and individuals with overweight or obesity (n = 5). Scale bars, 50 μm (top) and 20 μm (bottom). **i** Representative immunofluorescence images of staining with antibodies against CD68 (green), MVP (red), and Perilipin (purple) in visceral adipose tissue of overweight or obese individuals. Nuclei were stained with DAPI (blue). Scale bars, 20 μm. **j** mRNA level of MVP in CD14[+] macrophages isolated from the visceral adipose tissue SVFs of normal weight (18.5 ≤ BMI < 24, n = 15) and overweight (24 ≤ BMI < 28) or obese (BMI ≥ 28) (n = 24) subjects. Data are expressed as mean ± SEM. *P < 0.05 and **P < 0.01 by Student's t test or ANOVA with post hoc test

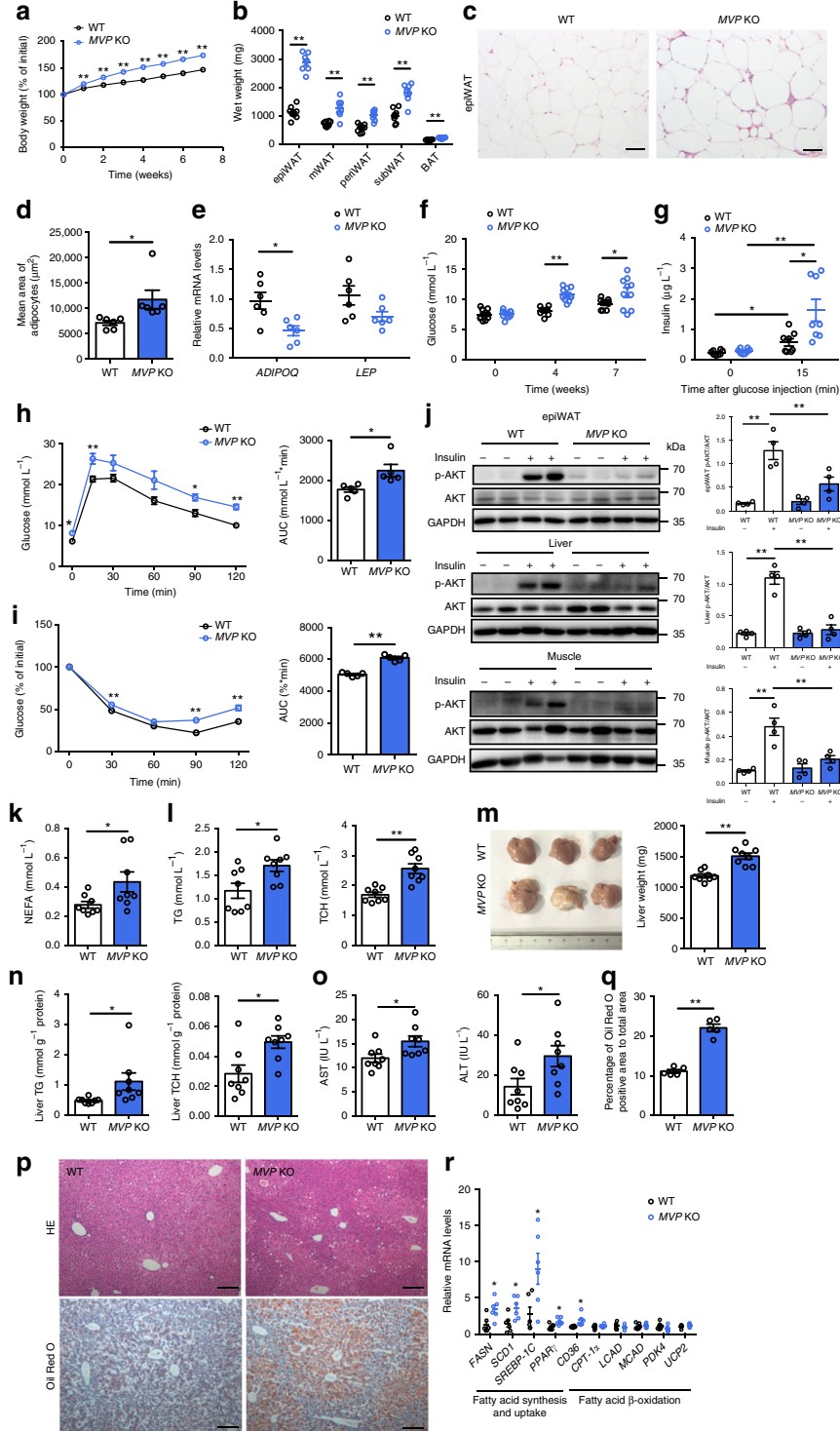

**Fig. 2** MVP deficiency deteriorates HFD-induced metabolic disorders in mice. Male WT and *MVP* KO mice were fed a HFD for 7 weeks. **a** The percentage of body weight gain in WT and *MVP* KO mice (*n* = 11). **b** Depot mass of epi, mesentery (m), perirenal (peri), subcutaneous (sub) WAT and BAT in WT and *MVP* KO mice (*n* = 8). **c** H&E staining of epiWAT from WT and *MVP* KO mice. Scale bars, 50 μm. **d** Quantification of adipocyte size in epiWAT of WT and *MVP* KO mice (*n* = 6). **e** mRNA levels of *ADIPOQ* and *LEP* in epiWAT from WT and *MVP* KO mice (*n* = 6). **f** Fasting blood glucose in WT and *MVP* KO mice (*n* = 10). **g** Basal- and stimulated-insulin levels in WT and *MVP* KO mice (*n* = 8). **h, i** GTT and ITT in WT and *MVP* KO mice (*n* = 5). **j** Western blot of AKT phosphorylation in the murine epiWAT, liver, and skeletal muscle stimulated by insulin. **k, l** Plasma levels of NEFA (**k**), TG and TCH (**l**) in WT and *MVP* KO mice (*n* = 8). **m, n** Murine liver tissues were retrieved after 7 weeks of HFD feeding and their wet weights (**m**), liver TG and TCH (**n**) levels were determined (*n* = 8). **o** Plasma levels of AST and ALT in mice (*n* = 8). **p** H&E (top) and Oil Red O (bottom) staining of representative liver sections obtained from HFD-fed WT (left) and *MVP* KO (right) mice. Scale bars, 100 μm. **q** Quantification of Oil Red O stained area of liver (*n* = 5). **r** mRNA levels of lipid metabolism-related genes in livers from WT and *MVP* KO mice (*n* = 6). Data are expressed as mean ± SEM. *P < 0.05 and **P < 0.01 by Student's *t* test or ANOVA with post hoc test

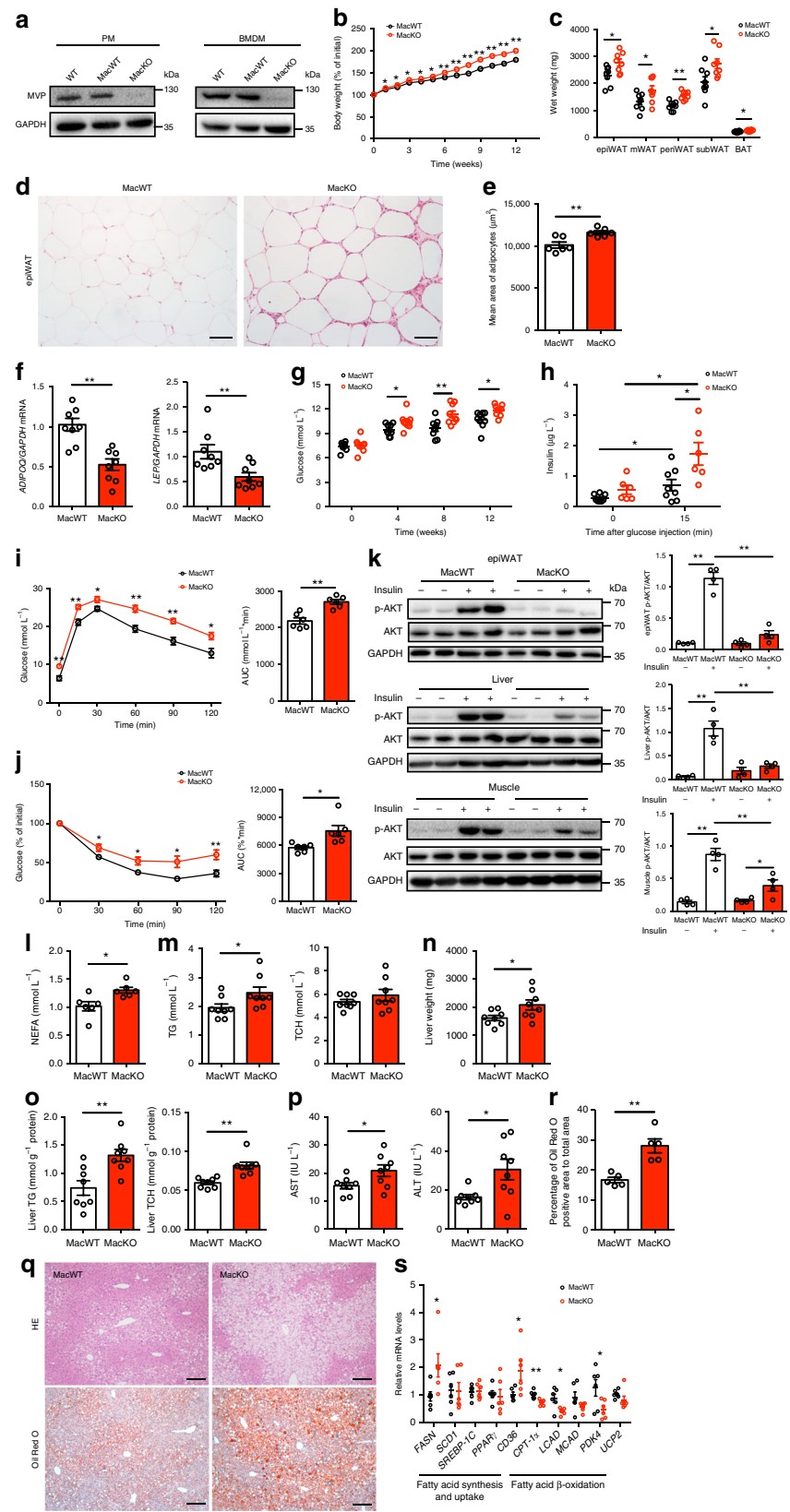

Fig. 3). Therefore, MVP deficiency may deteriorate HFD-induced obesity and obesity-associated metabolic disorders including insulin resistance, dysregulation of glycolipid metabolism, and liver steatosis in mice.

**Myeloid MVP deficiency exacerbates metabolic disorders.** Since MVP predominantly localized in macrophages in obese adipose tissues, we further generated a mouse model with myeloid-specific deletion of MVP (MacKO, *MVP*^flox/flox^*Lyz2*-Cre) by

**Fig. 3** Myeloid *MVP* deletion deteriorates obesity and metabolic disorders. MacWT and MacKO male mice were fed a HFD for 12 weeks. **a** Western blot analysis of MVP expression in PMs and BMDMs in WT, MacWT, and MacKO mice. **b** The percentage of body weight gain in MacWT ($n = 9$) and MacKO ($n = 10$) mice during 12 weeks of HFD feeding. **c** Depot mass of epi, m, peri, subWAT, and BAT in MacWT and MacKO mice ($n = 8$). **d** Histological analysis of epiWAT from MacWT and MacKO mice using H&E staining. Scale bars, 50 μm. **e** Quantification of adipocyte size in epiWAT of MacWT and MacKO HFD-fed mice ($n = 6$). **f** mRNA levels of *ADIPOQ* and *LEP* in epiWAT from MacWT and MacKO HFD-fed mice ($n = 8$). **g** Fasting blood glucose in MacWT and MacKO mice ($n = 9$). **h** Basal- and stimulated- insulin levels in MacWT ($n = 8$) and MacKO ($n = 6$) mice. **i, j** GTT and ITT in MacWT and MacKO mice ($n = 6$). **k** Western blot analysis of AKT phosphorylation in the murine epiWAT, liver, and skeletal muscle after insulin administration in vivo. **l, m** The NEFA (**l**) ($n = 6$), TG and TCH (**m**) ($n = 8$) contents in the plasma of MacWT and MacKO mice. **n, o** Murine liver tissues were retrieved and their wet weight (**n**), liver TG and TCH contents (**o**) were determined ($n = 8$). **p** Plasma levels of AST and ALT in MacWT and MacKO mice ($n = 8$). **q** H&E (top) and Oil Red O (bottom) staining of representative liver sections obtained from HFD-fed MacWT (left) and MacKO (right) mice. Scale bars, 100 μm. **r** Quantification of Oil Red O stained area of liver ($n = 5$). **s** mRNA levels of lipid metabolism-related genes in livers from the HFD-fed MacWT and MacKO mice ($n = 6$). Data are expressed as mean ± SEM. *$P < 0.05$ and **$P < 0.01$ by Student's *t* test or ANOVA with post hoc test

establishing *MVP*$^{\text{flox/flox}}$ mice that were crossed with *Lyz2*-Cre mice (Supplementary Fig. 4a–b) to investigate the role of macrophage MVP in the pathogenesis of metabolic disorders. The absence of MVP was detected in both bone marrow-derived macrophages (BMDMs) and PMs (Fig. 3a) but not in other tissues (Supplementary Fig. 4c). Similar to *MVP* KO mice, MacKO mice fed a normal chow diet showed minimal changes in body weight and glycolipid metabolism (Supplementary Fig. 5). However, challenge with 12 weeks of HFD feeding resulted in a greater body weight gain in MacKO mice compared to control (MacWT, *MVP*$^{\text{flox/flox}}$) littermates (Fig. 3b). Consistently, MacKO mice displayed higher levels of various adipose depots weight (Fig. 3c), larger average adipocyte size (Fig. 3d, e), decreased levels of *ADIPOQ* and *LEP* in the epiWAT (Fig. 3f). These results indicate that myeloid MVP deficiency, specifically in macrophages, may exacerbate HFD-induced obesity in mice.

MVP deletion in macrophages also exacerbated HFD-induced insulin resistance (Fig. 3g–k), hyperlipidaemia (Fig. 3l, m) and liver steatosis (Fig. 3n–r) in mice. The pro-steatotic effects by macrophage MVP deficiency were presumably attributed to up-regulation of fatty acid synthesis and uptake genes (*FASN*, *CD36*) and down-regulation of fatty acid β-oxidation genes (*CPT-1α*, *LCAD*, and *PDK4*) in the liver (Fig. 3s). These data clearly demonstrate that macrophage MVP deficiency exhibited similar phenotypic effects in mice as the global MVP knockout. MVP in macrophages may play an important role in antagonizing obesity and obesity-associated metabolic disorders.

**MVP deficiency aggravates metabolic inflammation**. To explore the mechanisms underlying the antagonizing effects of MVP on metabolic disorders, we examined the relationship between MVP deficiency and inflammation. IHC analysis revealed that CD68$^+$ macrophages in epiWAT were significantly increased in HFD-fed *MVP* KO mice compared with WT control group (Fig. 4a). FACS measurements showed that the adipose tissues from obese *MVP* KO mice contained more SVFs (Fig. 4b) and macrophages (Fig. 4c, d). The pro-inflammatory cytokines *TNF-α* and *IL-1β* and chemokines *CCL2* and *CCL3* were obviously increased in epiWAT, subWAT, BAT, and liver of HFD-fed *MVP* KO mice (Fig. 4e–h). The plasma levels of TNF-α and IL-1β were also increased consistently (Fig. 4j). Furthermore, the pro-inflammatory mediators were significantly increased in F4/80$^+$ macrophages isolated from epiWAT of HFD-fed *MVP* KO mice (Fig. 4i), suggesting that the macrophage be an important source for pro-inflammatory mediators in the obese *MVP* KO mice.

We further validated the role of macrophage MVP in inflammatory responses by using the MacKO mice models. As expected, the phenotypes displayed by the MacKO mice were similar to those of *MVP* KO mice in the obesity-induced inflammation (Fig. 4k–t). Taken together, our results suggest that

MVP may inhibit macrophage-orchestrated inflammatory responses in obese mice.

**MVP deficiency promotes atherosclerosis**. Atherosclerosis is also a metabolic disease in which inflammation is involved in the whole process of pathogenesis[4,7,18]. We found that MVP expression was up-regulated in the atherosclerotic plaques induced by western diet (WD) administered for 10 weeks to *ApoE* knockout (*ApoE*$^{\text{KO}}$) mice (Supplementary Fig. 6a–b). MVP was mainly expressed in CD68$^+$ macrophages in the mouse aortic roots (Supplementary Fig. 6c). In order to investigate the impact of MVP on atherosclerosis, we generated the *MVP* and *ApoE* double knockout (*MVP*$^{\text{KO}}$*ApoE*$^{\text{KO}}$) mice. After feeding the mice with a WD for 10 weeks, we did not observe significant difference in serum lipid levels between *MVP*$^{\text{KO}}$*ApoE*$^{\text{KO}}$ mice and *MVP*$^{\text{WT}}$*ApoE*$^{\text{KO}}$ littermates (Supplementary Fig. 6d). However, atherosclerotic lesion in the aorta was increased in *MVP*$^{\text{KO}}$*ApoE*$^{\text{KO}}$ mice in comparison with *MVP*$^{\text{WT}}$*ApoE*$^{\text{KO}}$ littermates (Fig. 5a, d). *MVP*$^{\text{KO}}$*ApoE*$^{\text{KO}}$ mice suffered from larger lesions (Fig. 5b, e) with more CD68$^+$ plaque area (Fig. 5c, f), suggesting that MVP deletion may promote atherosclerosis in mice.

To further understand the role of macrophage MVP in atherosclerosis, we generated myeloid-specific *MVP* deficiency and *ApoE* knockout mice (*MVP*$^{\text{MacKO}}$*ApoE*$^{\text{KO}}$, *MVP*$^{\text{flox/flox}}$*ApoE*$^{\text{KO}}$*Lyz2*-Cre) and the littermates (*MVP*$^{\text{MacWT}}$*ApoE*$^{\text{KO}}$, *MVP*$^{\text{flox/flox}}$*ApoE*$^{\text{KO}}$) by crossing *MVP*$^{\text{flox/flox}}$ mice with *Lyz2*-Cre mice and *ApoE*$^{\text{KO}}$ mice. After feeding on a WD for 12 weeks, *MVP*$^{\text{MacKO}}$*ApoE*$^{\text{KO}}$ mice exhibited similar atherosclerotic lesion characteristics (Fig. 5g–l) and plasma lipids levels (Supplementary Fig. 6e) as *MVP*$^{\text{KO}}$*ApoE*$^{\text{KO}}$ mice. These results reveal that the MVP deficiency, or predominantly MVP deficiency in macrophages, may be a promoter of atherogenesis in mice.

**MVP deficiency stimulates inflammation in atherosclerosis**. The observation that more CD68$^+$ macrophages in *MVP* KO atherosclerotic lesions compelled us to further investigate the role of MVP in macrophage accumulation. We found the Ly-6C$^{\text{hi}}$ pro-inflammatory monocytes labeled by fluorescent beads, representing the newly recruited monocytes[26,27], were dramatically increased in the *MVP*$^{\text{KO}}$*ApoE*$^{\text{KO}}$ atherosclerotic lesions (Fig. 5m, n). In addition, the total peritoneal cells and F4/80$^+$ macrophages elicited by thioglycollate, an inducer of inflammation, were also obviously increased in *MVP* KO mice (Fig. 5o, p). These results indicate that MVP deficiency may stimulate mono-macrophages recruitment to the artery wall.

The macrophage infiltration in the tissue will most likely elicit an inflammatory response. To validate it, we measured the expression levels of multiple inflammatory mediators in the

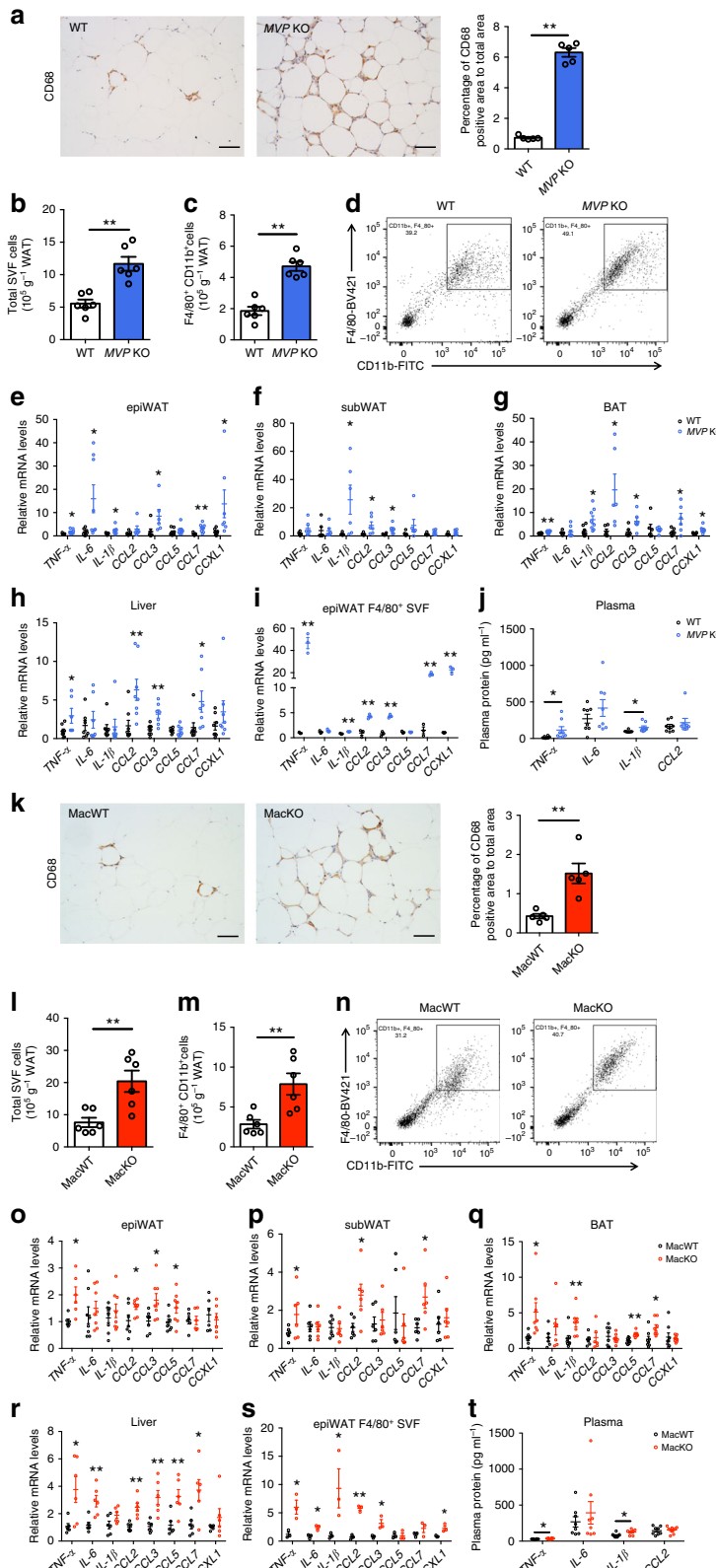

mouse aortic lesions. A robust increase in pro-inflammatory mediators such as *TNF-α*, *IL-6*, *IL-1β*, and *CCL2* were observed in *MVP*$^{KO}$*ApoE*$^{KO}$ mice and *MVP*$^{MacKO}$*ApoE*$^{KO}$ mice compared to their control littermates (Fig. 5q, r). Consistently, plasma levels of TNF-α, IL-1β, and CCL2 were also significantly increased in *MVP*$^{MacKO}$*ApoE*$^{KO}$ mice (Fig. 5s). Therefore, MVP deficiency

may result in vigorous inflammation in the artery wall. These results were further corroborated by in vitro experiments, in which administration of lipopolysaccharide (LPS) (Fig. 6a, b) but not TNF-α (Supplementary Fig. 7a) caused a dramatically increased production of TNF-α and CCL2 in the *MVP* KO PMs compared with controls.

**Fig. 4** MVP deficiency promotes inflammation in HFD-fed mice. **a** Representative CD68[+] staining in epiWAT from HFD-fed WT and *MVP* KO mice ($n = 5$). Scale bars, 50 μm. **b**, **c** Quantification of epiWAT SVFs (**b**) and macrophages (**c**) by flow cytometry in HFD-fed WT and *MVP* KO mice ($n = 6$). **d** Representative flow cytometry plot charts of F4/80[+]CD11b[+] macrophages in epiWAT of HFD-fed WT and *MVP* KO mice. **e–i** mRNA levels of inflammatory mediators in epiWAT (**e**), subWAT (**f**), BAT (**g**), liver (**h**) ($n = 6-8$), and epiWAT F4/80[+] macrophages ($n = 3$) (**i**) in HFD-fed WT and *MVP* KO mice. **j** Plasma concentrations of TNF-α, IL-6, IL-1β, and CCL2 in HFD-fed WT and *MVP* KO mice ($n = 8$). **k** Representative CD68[+] staining in epiWAT from HFD-fed MacWT and MacKO mice ($n = 5$). Scale bars, 50 μm. **l**, **m** Quantification of epiWAT SVFs (**l**) and macrophages (**m**) by flow cytometry in HFD-fed MacWT and MacKO mice ($n = 6$). **n** Representative flow cytometry plot charts of F4/80[+]CD11b[+] macrophages in epiWAT of HFD-fed MacWT and MacKO mice. **o–s** mRNA levels of inflammatory mediators in epiWAT (**o**), subWAT (**p**), BAT (**q**), liver (**r**) ($n = 6-8$), and epiWAT F4/80[+] macrophages ($n = 3$) (**s**) in HFD-fed MacWT and MacKO mice. **t** Plasma concentrations of TNF-α, IL-6, IL-1β, and CCL2 in HFD-fed MacWT and MacKO mice ($n = 8$). Data are expressed as mean ± SEM. *$P < 0.05$ and **$P < 0.01$ by Student's *t* test

**MVP deficiency activates NF-κB signaling in macrophages**. NF-κB is a key transcription factor governing the expression of most pro-inflammatory genes[15]. To understand how MVP modulates inflammatory responses in macrophages, we examined the relationship between MVP and NF-κB signaling pathway. Figure 6c, d shows that the loss of MVP strongly stimulated the phosphorylation of IKK and p65 and IκBα degradation, which led to the translocation of the NF-κB complex into the nucleus to initiate transcription. Indeed, we observed an enhanced nuclear translocation of p65 in *MVP* KO macrophages (Fig. 6e–g). Furthermore, when macrophages were treated with Bay11-7082, an inhibitor of IκBα, MVP deficiency induced over-production of *TNF-α* and *CCL2* in PMs was effectively reversed (Fig. 6h). Consistently, MVP deficiency also increased macrophage p-p65 in the mouse epiWAT (Supplementary Fig. 8a). Moreover, the activated degree of NF-κB signaling and overproduction of inflammatory cytokines were much stronger than that of the MVP up-regulation in the obese murine epiWAT macrophages (Supplementary Fig. 8b-c). In the overweight/obese human subjects, the expression of *MVP* was negatively correlated with *CCL2* in the visceral adipose tissue macrophages (Fig. 6i). These data suggest that MVP deficiency or insufficient expression may stimulate inflammatory response by activating NF-κB signaling pathway in macrophages.

Next, we sought to determine the molecular mechanisms by which MVP modulates NF-κB signaling pathway in macrophages. We first tested whether MVP might directly interact with p65. Co-immunoprecipitation (Co-IP) experiments showed that MVP did not form a complex with p65 (Supplementary Fig. 9a). Secondly, we investigated the potential molecular link between MVP and TRAF6, a key regulator in the activation of NF-κB[14]. After cell fractionation by ultracentrifugation, TRAF6 but not TRAF2 or TRAF3 could be detected in the macrophage vault pellet, reflecting that the assembled MVP but not the free MVP may interact with TRAF6 (Supplementary Fig. 9b-c). Co-IP revealed an interaction between endogenous MVP and TRAF6 in macrophages (Fig. 6j). When both Flag-tagged MVP and HA-tagged TRAF6 were co-transfected into HEK293T cells, Flag-MVP was detected mainly in the pellet but not in the supernatant after cell fractionation, indicating that Flag-MVP may exist as the assembled vaults in cells. HA-TRAF6 could be co-precipitated with Flag-MVP in the pellet (Supplementary Fig. 9d). Co-IP with Flag or HA antibody also showed that Flag-MVP directly interacted with HA-TRAF6 in cells (Fig. 6k). Upon LPS stimulation, more TRAF6 was co-precipitated with the assembled MVP in macrophages after cell fractionation (Supplementary Fig. 9e) and co-IP with TRAF6 antibody (Fig. 6l). Furthermore, the MVP–TRAF6 complex formation was enhanced in the obese murine epiWAT SVFs (Supplementary Fig. 9f). Immunofluorescence staining showed that MVP predominantly co-localized with TRAF6 in the cytoplasm of PMs (Fig. 6m).

**MVP inhibits the polyubiquitination of TRAF6 in cells**. TRAF6 polyubiquitination is a key step in the NF-κB signaling pathway. Upon LPS stimulation, the E3 ligase activity of TRAF6 is induced and the activated TRAF6 targets itself and other molecules for polyubiquitination[10,14]. MVP depletion significantly enhanced LPS-induced polyubiquitination of TRAF6 in murine BMDMs (Fig. 7a). In contrast, the overexpression of MVP strongly inhibited TRAF6 polyubiquitination in HEK293T cells (Fig. 7b). Therefore, MVP may prevent NF-κB activation via inhibition of TRAF6 polyubiquitination.

TRAF6 polyubiquitination depends on its recruitment to IRAK1 and subsequent oligomerization[28–30]. We found that MVP deficiency increased TRAF6 recruitment to IRAK1 in BMDMs upon the LPS stimulation (Fig. 7c). Overexpression of MVP inhibited the complex formation of IRAK1 with TRAF6 (Fig. 7d). Furthermore, the presence of MVP prevented the TRAF6 oligomerization (Fig. 7e) while IRAK1 promoted TRAF6 oligomerization (Supplementary Fig. 10a) in cells. TRAF6 is composed of an amino (N)-terminal RING-finger domain, several zinc-finger domains, and a conserved carboxy (C)-terminal TRAF domain[30,31]. Considering the structural features of TRAF6, we generated two truncated fragments of TRAF6 (Supplementary Fig. 10b) both carrying the HA tag. Co-IP showed that MVP recognized both fragments of TRAF6 (Supplementary Fig. 10c). However, only the C-terminal fragment (332–530) interacted with IRAK1 (Supplementary Fig. 10d). This is consistent with the concept that TRAF-C terminal domain is responsible for the interaction of TRAF6 with IRAK1 and other signaling molecules[13,29,32].

To understand how MVP exerts its inhibitory effects on TRAF6, we further generated three truncated fragments of MVP with a Flag tag (Supplementary Fig. 10b). All three expressed truncates in HEK293T cells existed in both the supernatant and pellet after cell fractionation, while the full-length MVP (MVP-FL) was mostly detected in the pellet (Supplementary Fig. 10e). These three truncated MVPs could bind with TRAF6 (Supplementary Fig. 10f). However, only MVP-FL and MVP α-helical domain (686–870) could substantially block the oligomerization (Fig. 7f) and the self-ubiquitination (Fig. 7g) of TRAF6 simultaneously. Accordingly, the overexpression of MVP-FL and MVP-(686–870) strongly inhibited the LPS-induced (Fig. 7h–j) but not the TNF-α-induced production of inflammatory cytokines (Supplementary Fig. 11a) and the nuclear translocation of p65 (Fig. 7k) in macrophages. MVP-(1–480) and MVP-(481–685) did not influence the LPS-induced inflammatory cytokines production in cells (Supplementary Fig. 11b-c). As such, our data reveal that MVP suppresses inflammatory responses by specifically binding to TRAF6 and preventing TRAF6 oligomerization and ubiquitination in macrophages.

**Discussion**

Chronic inflammation is a common feature of obesity and atherosclerosis, and contributes greatly to the pathogenesis of metabolic

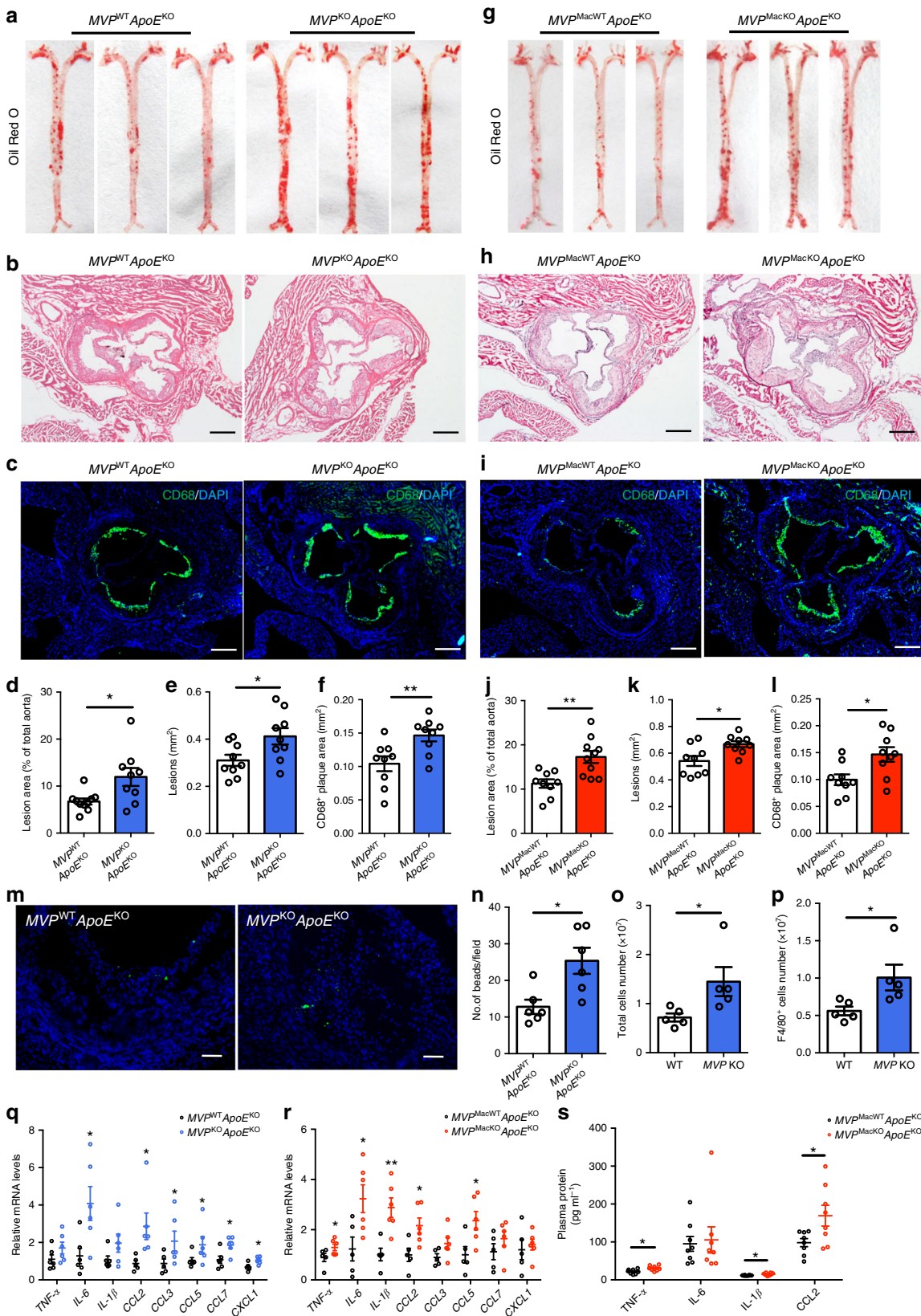

diseases. Vast pharmacological efforts have been invested in developing treatments for metabolic diseases by focusing on pro-inflammatory cytokines as TNF-α, IL-1β, and IL-6[1,2,33,34]. However, these approaches had limited success. The difficulty in translation underscores the complexity of the metabolic

inflammation in the body and highlights a huge gap in the understanding of the mechanisms underlying metabolic diseases. In particular, the intrinsic regulatory elements in inflammatory pathways may fulfill an equally critical role in the immunometabolic homeostasis. In the present study, we have provided a critical proof

**Fig. 5** Deficiency of MVP accelerates atherosclerosis progression. **a**, **d** En face Oil Red O staining of whole aortas from $MVP^{KO}ApoE^{KO}$ ($n = 9$) and control $MVP^{WT}ApoE^{KO}$ ($n = 10$) male mice fed with a WD for 10 weeks (**a**). Lesion occupation was quantified and shown in (**d**). **b**, **e** Representative H&E-stained images (**b**) and quantitative analysis (**e**) of the lesions in aortic root sections from $MVP^{KO}ApoE^{KO}$ and $MVP^{WT}ApoE^{KO}$ mice ($n = 9$). Quantification of lesion burden was performed by cross-sectional analysis of the aortic root. Scale bars, 200 µm. **c**, **f** Representative CD68$^+$ staining in cross-sections (**c**) and quantitative analysis (**f**) of the aortic root plaques from $MVP^{KO}ApoE^{KO}$ and $MVP^{WT}ApoE^{KO}$ mice ($n = 9$). Scale bars, 200 µm. **g**, **j** En face Oil Red O staining of aortas from $MVP^{MacKO}ApoE^{KO}$ ($n = 10$) and control $MVP^{MacWT}ApoE^{KO}$ ($n = 9$) mice fed a WD for 12 weeks (**g**). Lesion occupation was quantified and shown in (**j**). **h**, **k** Representative H&E-stained images (**h**) and quantitative analysis (**k**) of the lesions in aortic root sections from $MVP^{MacKO}ApoE^{KO}$ and $MVP^{MacWT}ApoE^{KO}$ mice ($n = 9$). Scale bars, 200 µm. **i**, **l** Representative CD68$^+$ staining in cross-sections (**i**) and quantitative analysis (**l**) of the aortic root plaques from $MVP^{MacKO}ApoE^{KO}$ and $MVP^{MacWT}ApoE^{KO}$ mice ($n = 9$). Scale bars, 200 µm. **m**, **n** Quantitative analysis of infiltrated fluorescent bead-labeled monocytes in atherosclerotic lesions of $MVP^{KO}ApoE^{KO}$ and $MVP^{WT}ApoE^{KO}$ mice fed with a WD for 10 weeks ($n = 6$). **o**, **p** Three days after intraperitoneal injection of 1 ml 4% sterile thioglycollate media, total number of peritoneal cells (**o**) and F4/80$^+$ PMs (**p**) of WT and $MVP$ KO mice were measured ($n = 5$). **q**, **r** mRNA levels of inflammatory mediators in the aortas of $MVP^{KO}ApoE^{KO}$ (**q**) and $MVP^{MacKO}ApoE^{KO}$ (**r**) mice ($n = 5-6$). **s** Plasma concentration of TNF-α, IL-6, IL-1β, and CCL2 in $MVP^{MacKO}ApoE^{KO}$ and $MVP^{MacWT}ApoE^{KO}$ mice ($n = 8$). Data are expressed as mean ± SEM. *$P < 0.05$ and **$P < 0.01$ by Student's $t$ test

of principle that MVP, the major component of vaults, may act as an intrinsic inflammatory gatekeeper in macrophages to regulate obesity-associated metabolic disorders and atherosclerosis.

Obesity facilitates the development of many metabolic disorders. On the contrary, weight reduction, achieved through bypass surgery or otherwise, confers effective therapeutic benefit. This may also contribute to the underlying core mechanism of MVP antagonizing insulin resistance, hyperlipidaemia, and liver steatosis in mice. In addition, we provide evidence to show that macrophage MVP is the major source of the anti-obesity and anti-inflammatory signal curbing the development of metabolic disorders, because MVP was up-regulated primarily in macrophages and specific deletion of MVP in macrophages sufficed to aggravate HFD-induced obesity in mice. This unique feature of MVP separates it from other obesity-associated molecules like adipocyte fatty-acid-binding protein aP2, which integrates metabolic and inflammatory responses in both adipocytes and macrophages[35]. However, the observation that the pro-obesity effect of MVP deficiency in macrophages was somewhat less prominent compared to that of global MVP deficiency in mice suggests that other sources of MVP may contribute to the early phase of weight-reducing action. MVP has been shown to be expressed and functional in endothelial cells and hepatocytes[22,25]. The role of other cell and tissue sources of MVP in obesity warrants to further investigation.

The macrophage-autonomous MVP may be critically involved in suppressing the magnitude and duration of metabolic inflammation. This conclusion is supported by three key findings though the pro-inflammatory feature of MVP has been reported in certain situations[36,37]: First, MVP deficiency caused obvious macrophage infiltration in obese adipose tissues and in atherosclerotic lesions in mice. Second, pro-inflammatory chemokines and cytokines were dramatically increased in major metabolic tissues, in the circulation, and in atherosclerotic lesions of MVP deficiency mice. Third, the loss of MVP strongly activated the NF-κB signaling pathway in macrophages. In the TLR mediated inflammatory signaling pathway, IRAK1 activated by the MyD88-dependent pathway recruits TRAF6, promotes its oligomerization and complex formation with TAB2, TAK1, etc., to undergo polyubiquitination, thereby activating downstream IKK–NF-κB[11,13,14,29,38]. MVP exists and functions as the assembled macromolecular vault particle in cells[19]. MVP may bind directly with TRAF6, which is different from the interaction between IRAK1 and TRAF6 that promotes TRAF6 oligomerization and subsequent ubiquitination[14,28,29]. MVP seems to interact with all three domains of TRAF6, while IRAK1 interacts only with the TRAF-C domain[13,28]. The RING-finger and zinc-finger domains are requisite for the oligomerization and ubiquitination of TRAF6[29,32,39]. The unique binding pattern of MVP to TRAF6

impairs the oligomerization and ubiquitination of TRAF6. Thus, MVP may inhibit the IKK–NF-κB signaling by preventing IRAK1-induced TRAF6 oligomerization and ubiquitination in macrophages. Yet further studies are needed to elucidate the detailed molecular mechanisms.

As a suppressor of IKK–NF-κB signaling, it is intrigued that MVP expression is induced in murine and human macrophages after the onset of obesity. The observed up-regulation of MVP in obesity-associated metabolic disorders and atherosclerotic lesions may be elicited by inflammation. The promoter of MVP contains binding sites for some important pro-inflammatory transcription factors such as SP1 and STAT1[21,40]. Obesity may induce an insufficient up-regulation of MVP comparing with a strong induction of inflammatory response in the body. Moreover, the enhanced MVP levels associate with TRAF6 thereby inhibiting its activation and consequently suppressing NF-κB signaling. All the three distinct protein domains of MVP could bind to TRAF6. The α-helical domain of MVP is crucial for the interaction between MVP molecules and vault assembly[41]. We demonstrate that TRAF6 binding to this domain is instrumental in preventing the oligomerization and ubiquitination of TRAF6. MVP seems not to interact with TRAF2 or TRAF3. It may not influence the TNFα-induced pro-inflammatory cytokines production in macrophages. The selective inhibition of NF-κB up-stream signaling reveals that MVP may be unable to suppress metabolic inflammation completely. This may partly explain the result that the MVP expression was negatively correlated with CCL2 but not with TNF-α in obese human macrophages. Thus, MVP may constitute an essential constraint in a negative feedback loop to fine-tune inflammatory responses in macrophages, that may contribute to "low grade and chronic" metabolic inflammation.

The role of IKK–NF-κB signaling in metabolic diseases is still a controversial issue. Although a detrimental role of IKK–NF-κB activation has been documented in multiple tissues, there have been conflicting results that cannot be neglected. For example, IKKβ is considered essential in the regulation of adipocyte survival and adaptive remodeling in obese mice[42]. In addition, the IKK–NF-κB pathway can potentially dampen rather than instigate inflammation through anti-inflammatory cytokine production in the adipose tissue and artery[43,44]. Our study demonstrated that excessive input of nutrition could activate IKK–NF-κB signaling pathway and inflammation in macrophages, which was strongly attenuated by MVP. Upstream regulators like MVP may influence the activity of IKK that would activate NF-κB signaling. Consistently, leukocyte immunoglobulin-like receptor B4 (LILRB4) recruits SHP1 for inhibiting TRAF6 ubiquitination and subsequently inactivating NF-κB cascades to attenuate nonalcoholic fatty liver disease[45]. Conceivably, the autonomous negative regulation of TRAF6 by different factors including MVP may

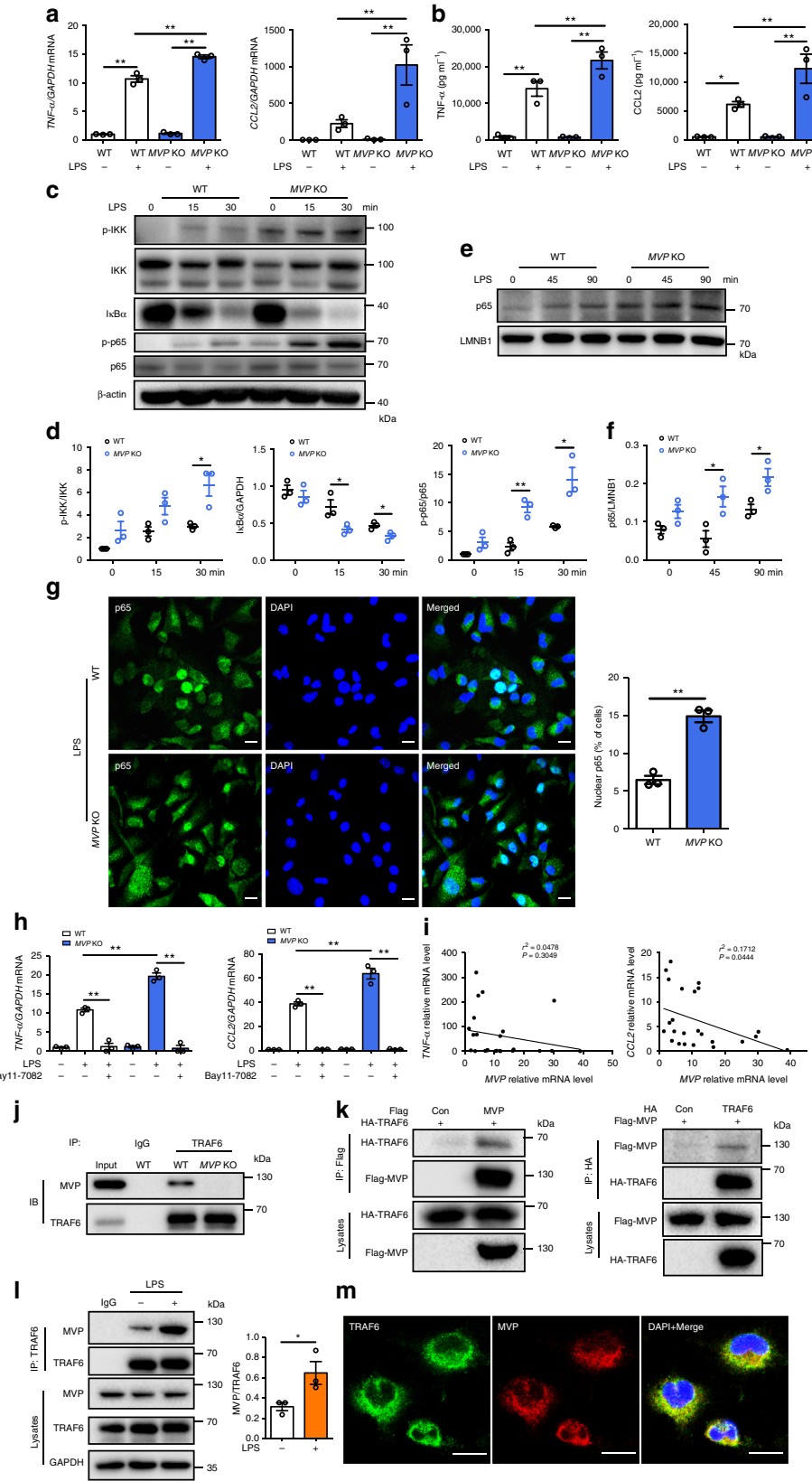

determine the precise role of IKK and as such the outcomes in the metabolic diseases.

In summary, our findings demonstrate that the macrophage MVP functions as a crucial constraint for metabolic inflammation, in which it attenuates obesity-associated metabolic disorders and atherosclerosis. Identification of autonomous regulatory mechanism is of special importance for understanding the nature of inflammatory response. This will hopefully open the door to the development of more effective intervention strategies for the metabolic diseases.

**Fig. 6** MVP deficiency activates IKK–NF-κB signaling. PMs isolated from CD-fed WT and *MVP* KO mice were treated with LPS (100 ng ml$^{-1}$) for indicated times. **a**, **b** PMs were stimulated with LPS for 12 h and mRNA levels of inflammatory mediators (*TNF-α* and *CCL2*) were assessed by RT-qPCR (*n* = 3) (**a**). TNF-α and CCL2 levels in culture media were determined using ELISA (*n* = 3) (**b**). **c**, **d** Western blot analysis of p-IKK, IKK, IκBα, p-p65, and p65 in PMs that were treated with LPS for indicated times (*n* = 3). **e**, **f** Western blot analysis of nuclear extracts prepared from PMs stimulated with LPS for the indicated times and analyzed for p65 and Lamin B1 (*n* = 3). **g** Representative immunofluorescence images of p65 nuclear translocation assay. PMs were stimulated with LPS for 3 h and analyzed for p65 localization by immunofluorescence staining. Nuclei were stained with DAPI, and the percentage of nuclear p65 positive cells was counted. Scale bars, 10 μm. **h** mRNA levels of inflammatory mediators in PMs cultured with or without LPS or NF-κB pathway inhibitor BAY11-7082 (*n* = 3). **i** Correlative analysis of the expression of *MVP* versus *TNF-α* and *CCL2* in CD14$^+$ macrophages from the visceral adipose tissue of overweight/obese human subjects (*n* = 24). **j** Co-IP and western blot analysis of endogenous MVP and TRAF6 from protein lysates of murine BMDMs. **k** HEK293T cells were transfected with control Flag empty vector or Flag-MVP and HA-TRAF6 plasmids (left), and control HA empty vector or HA-TRAF6 and Flag-MVP plasmids (right). Co-IP and western blot analysis with anti-HA and anti-Flag antibodies. **l** Co-IP and quantification of the interaction between MVP and TRAF6 in response to LPS stimulation in BMDMs. **m** Representative immunofluorescence images of PMs stained by anti-MVP (red) and anti-TRAF (green) antibodies to examine the distribution of MVP and TRAF6. Scale bars, 10 μm. Representative results from three independent experiments are shown. Data are expressed as mean ± SEM. *P < 0.05 and **P < 0.01 by Student's *t* test or ANOVA with post hoc test

## Methods

**Mice**. *MVP*$^{flox/flox}$ mice were generated by Shanghai Model Organisms Center, Inc. (Shanghai, China), using a targeting vector generated by ET cloning techniques. In this vector, a neomycin selection cassette flanked by two Frt sites with a loxP site was inserted into the upstream of exon 2 of the targeted gene. Another loxP site was inserted into the downstream of exon 3 (Supplementary Fig. 4a). The targeting vector was electroporated into C57BL/6 Bruce4 embryonic stem (ES) cells. The correctly recombined ES colony was then injected into C57BL/6 blastocysts. Male chimeras were mated with female C57BL/6 mice to get mice with a targeted *MVP* allele. The mice were crossbred with C57BL/6 flp-recombinase mice to remove the neomycin cassette to create heterozygous *MVP*$^{flox/+}$ mice. The mice were then crossbred with C57BL/6 mice for nine generations before being bred with heterozygous *MVP*$^{flox/+}$ mice to get the *MVP*$^{flox/flox}$ mice. One set of primers were used to genotype the mice by PCR on DNA isolated from tails (forward 5′-CACAGTGCACATAAACTTATGCAA and reverse 5′-TGATGTTCCAAAGGA-GACAGTAAA), resulting in an 895-bp fragment in *MVP*$^{flox/flox}$ mice and a 771-bp fragment in WT mice.

To generate myeloid-specific MVP deficient mice, *MVP*$^{flox/flox}$ mice were crossed with a C57BL/6J mouse expressing Cre recombinase from the *Lyz2* promoter (B6.129P2-*Lyz2*$^{tm1(cre)Ifo}$/J), which termed as MacKO (*MVP*$^{flox/flox}$*Lyz2*-Cre) mice. Mice containing the floxed *MVP* allele that did not express the Cre recombinase gene (*MVP*$^{flox/flox}$) were used as the control (termed as MacWT mice).

For atherosclerosis experiments, *MVP* KO mice were subsequently bred with apolipoprotein E-deficient (*ApoE*$^{-/-}$, *ApoE*$^{KO}$) mice (B6.129P2-*Apoe*$^{tm1Unc}$/J) to generate *MVP*$^{-/-}$*ApoE*$^{-/-}$ (*MVP*$^{KO}$*ApoE*$^{KO}$) mice and their littermates *ApoE*$^{-/-}$ (*MVP*$^{WT}$*ApoE*$^{KO}$) mice. For generating myeloid-specific *MVP* deficient mice in an *ApoE*$^{-/-}$ background, *MVP*$^{flox/flox}$ and *Lyz2*-Cre mice were firstly backcrossed onto the *ApoE*$^{-/-}$ mice. *ApoE*$^{-/-}$*MVP*$^{flox/flox}$ mice were then crossed with *ApoE*$^{-/-}$*Lyz2*-Cre mice to generate *ApoE*$^{-/-}$*MVP*$^{flox/flox}$*Lyz2*-Cre mice, which were termed as *MVP*$^{MacKO}$*ApoE*$^{KO}$ mice. *ApoE*$^{-/-}$*MVP*$^{flox/flox}$ mice (termed as *MVP*$^{MacWT}$*ApoE*$^{KO}$ mice) were used as controls.

**Animal models**. Mice were housed at 22–24 °C under standard light conditions (12 h light/dark cycle) and were allowed free access to water and food. For HFD-induced obesity model, experimental 7–8-week-old KO mice and their control mice were fed with either a normal CD or a HFD that contained 60% of its calories from fat (D12492, Research Diets) for 7 or 12 weeks. Body weight and blood glucose were measured weekly. For atherosclerosis experiments, experimental male mice aged 7–8-week-old KO mice and their control mice were fed with a WD that contained 1.25% cholesterol (D12108C, Research Diets) for 10 or 12 weeks. All animal protocols were approved by the Institutional Animal Care and Use Committee of Nanjing Medical University. All relevant ethical regulations were adhered to.

**Human tissue samples**. Tissue biopsies from visceral adipose tissue, obtained during surgery, were stored at −80 °C until further processing. All subjects provided their written informed consent. All procedures that involved human samples were approved by the Ethics Committee of Bayi Clinical Medicine School of Nanjing Medical University. All relevant ethical regulations were followed. To examine MVP expression, paraffin sections were stained with an anti-MVP antibody (Santa Cruz, sc-18701, 1:50). To isolate SVFs, human adipose tissues were digested using collagenase type II (1.5 mg ml$^{-1}$, Sigma) at 37 °C for 40 min. After passing cells through a 200 μm cell strainer and centrifugation at 1000*g* for 10 min, the pellet containing the SVFs was then incubated with red blood cell lysis buffer. SVFs were resuspended in phosphate-buffered saline (PBS) supplemented with 1% fetal bovine serum (FBS, Gibco). CD14$^+$ macrophages were purified using magnetic beads (BD Biosciences), according to the manufacturer's instructions. Cells were immediately used for total RNA extraction.

**Cell culture**. Primary mouse PMs and BMDMs were isolated and maintained as described[46,47]. PMs were harvested from the peritoneal cavity, washed with PBS, resuspended in Roswell Park Memorial Institute (RPMI, Gibco) 1640 medium containing 10% (v/v) FBS, supplemented with 1% penicillin/streptomycin (P/S). After 2 h incubation at 37 °C, nonadherent cells were removed, and the remaining adherent cells were cultured. To isolate BMDMs, 3–4-week-old mice were euthanized, and their femurs and tibias were collected. Bone marrow cells were cultured and differentiated for 7 days in RPMI 1640 medium supplemented with 10% FBS, 1% P/S, and 20 ng ml$^{-1}$ M-CSF (Sigma-Aldrich). Cells were treated with 100 ng ml$^{-1}$ lipopolysaccharide (LPS, Sigma-Aldrich) or 10 ng ml$^{-1}$ TNF-α (R&D Systems) for indicated times for analysis. RAW264.7 and HEK293T cells (ATCC) were cultured in Dulbecco's Modified Eagle's Medium (DMEM) supplemented with 10% FBS and 1% P/S.

**Intraperitoneal glucose and insulin tolerance tests**. Following an overnight fast, about 16 h, mice were intraperitoneally injected with glucose (1.5 g kg$^{-1}$), and blood samples for glucose determination were collected from the tail vein at the indicated times. Insulin tolerance was assessed after a 6 h fast by intraperitoneal injection of human regular insulin (1 U kg$^{-1}$) and blood glucose monitoring. Glycemia was assessed using the OneTouch Horizon Glucose Monitoring kit (LifeScan).

**In vivo insulin signaling**. For examination of in vivo insulin signaling, mice were fasted for 6 h, i.p. injected with human regular insulin (1 U kg$^{-1}$). Subsequently, mice were anesthetized and euthanized, and epiWAT, liver, and skeletal muscle were collected at the indicated times, flash-frozen in liquid nitrogen and stored at −80 °C until for western blot analysis with antibodies against phosphorylated AKT and total AKT.

**Analysis of metabolic parameters**. Blood glucose levels were measured using the OneTouch Horizon Glucose Monitoring kit (LifeScan) via tail vein blood sampling. Plasma insulin level in mice was measured using an insulin ELISA kit (Mercodia, Sweden). Plasma nonesterified fatty acids (NEFA), triglycerides (TG), total-cholesterol (TCH), low-density lipoprotein cholesterol (LDL-C), high-density lipoprotein cholesterol (HDL-C), AST, ALT and liver TG, TCH concentration were measured by using the enzymatic assays according to the manufacturer's instructions (Jiancheng Bio, China). Plasma TNF-α, IL-6, IL-1β, and CCL2 (eBioscience) concentration were determined by ELISA.

**Quantification of atherosclerosis burden**. Mice were euthanized and perfused with PBS through the left ventricle. Hearts and aortas were removed carefully and fixed with 4% paraformaldehyde. For en face analysis, the entire aorta was opened longitudinally, stained with Oil Red O, then placed on a blank sheet of paper and photographed with a Canon camera (PowerShot G12). Percentage of Oil Red O positive area was calculated using ImagePro Plus software. Hearts were dissected from the aorta and embedded in Tissue-Tek OCT compound (Sakura Finetek). For morphology analysis, aortic roots were cut in 5 μm-thick serial cryosections beginning from the onset of the aortic valves until the valves disappeared. Sections, each 80–100 μm apart, were mounted on one slide. Lesion size was quantified after H&E staining and calculated as the averages of 3 independent sections using ImagePro Plus software. Samples which exhibited evidence of artefactual tissue damage or abnormal orientation that could not be compensated by the analysis of multiple independent sections were excluded from analysis.

**Monocyte recruitment assays**. For the monocyte infiltration into atherosclerotic lesion assay, experimental male mice were fed a WD for 10 weeks. Clodronate-liposomes (250 μl, Liposoma) were i.v. injected in order to transiently deplete monocytes, followed by i.v. injection of 250 μl fluorescent microspheres 48 h later.

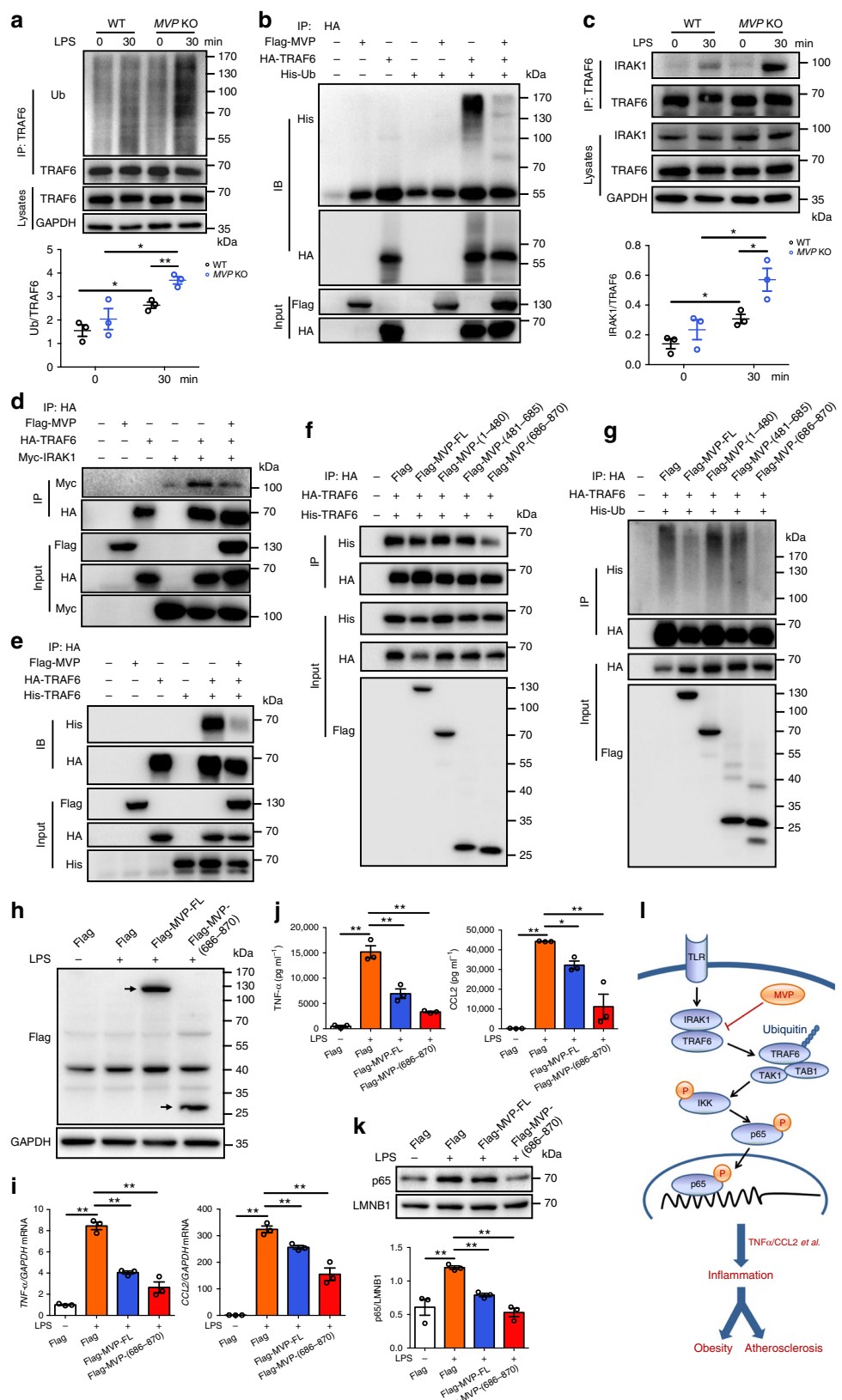

Fluoresbrite FITC-dyed (YG, 0.5 μm) plain microspheres (2.5% solids [w/v]; Polysciences) were diluted 1:25 in PBS[26,27]. Mice were euthanized and hearts with aortic root was then used for consecutive sections from the atrioventricular valve at a thickness of 20 μm. Nuclei were counter-stained by DAPI Fluor mount-G (SouthernBiotech). Images were then captured using a fluorescence microscope

(Carl Zeiss). Beads that reflect monocyte recruitment were quantified in 3–5 aortic sinus sections per mouse.

For the murine peritoneal mono-macrophage recruitment, 1 ml of sterile 4% thioglycolate media was injected intra-peritoneally. The cells from murine peritoneal cavities were harvested 3 days later, and analyzed by cell counter or flow cytometry.

**Fig. 7** MVP inhibits the activity of TRAF6 in cells. **a**, **b** Western blot analysis of the effect of MVP on TRAF6 ubiquitination in BMDMs (**a**) and HEK293T cells (**b**). **c**, **d** Western blot analysis of the effect of MVP on IRAK1–TRAF6 interaction in BMDMs (**c**) and HEK293T cells (**d**). **e** Western blot analysis of the effect of MVP on the oligomerization of TRAF6 in HEK293T cells. **f**, **g** Western blot analysis of the effect of full-length and truncated MVPs on the oligomerization (**f**) and ubiquitination (**g**) of TRAF6 in HEK293T cells. **h**–**k** RAW264.7 cells were transfected with lenti-viruses expressing control Flag, Flag-MVP-FL, and Flag-MVP-(686–870). Western blot analysis of Flag-MVP-FL and Flag-MVP-(686–870) expression in RAW264.7 cells (**h**). Expression of TNF-α and CCL2 in RAW264.7 cells treated by LPS. Expressional levels were measured by RT-qPCR (**i**) and ELISA (**j**) ($n = 3$). Western blot analysis of nuclear extracts prepared from RAW264.7 cells treated by LPS for 1 h (**k**). **l** Model illustrating the mechanism of MVP function in macrophages mediated metabolic inflammation in the context of obesity and atherosclerosis. Representative results from three independent experiments are shown. Data are expressed as mean ± SEM. *$P < 0.05$ and **$P < 0.01$ by ANOVA with post hoc test

**Histological analysis**. Formalin-fixed, paraffin-embedded tissue sections were routinely stained with H&E for the evaluation of the tissue morphology. The frozen liver sections were prepared using Tissue-Tek OCT compound (Sakura Finetek) and subjected to Oil Red O staining (Sigma-Aldrich) to visualize lipid droplets. The histological features were observed and captured under a light microscope (Carl Zeiss).

For immunohistochemical (IHC) staining, sections were incubated with primary antibodies against MVP (Santa Cruz, sc-18701, 1:50) and CD68 (Bio-Rad, MCA1957, 1:100) followed by incubation with the secondary antibodies conjugated with horseradish peroxidase. The sections were then treated with the ABC staining system (Santa Cruz) according to the instructions of the manufacturer. For all sections, 3,3-diaminobenzidine was used as the indicator substrate, which appeared as a brown reaction product. For immunofluorescence analysis, anti-MVP (goat, Santa Cruz, sc-18701, 1:50), anti-CD68 (rat, Bio-Rad, MCA1957, 1:100), anti-perilipin (rabbit, Cell Signaling, 9349, 1:100), anti-p-p65 (rabbit, Cell Signaling, 3033, 1:100), anti-p65 (rabbit, Cell Signaling, 8242, 1:100), and anti-TRAF6 (rabbit, Absin, abs115194, 1:100, China) antibodies were applied. The secondary antibodies were Alexa Fluor 546 donkey anti-goat IgG, Alexa Fluor 488 donkey anti-rat IgG, Alexa Fluor 647 donkey anti-rabbit IgG, Alexa Fluor 546 donkey anti-rabbit IgG, and Alexa Fluor 488 donkey anti-rabbit IgG (Thermo Fisher Scientific). Nuclei were counter-stained by DAPI Fluor mount-G (SouthernBiotech). Images were then captured using a fluorescence microscope (Carl Zeiss) or confocal microscope (Carl Zeiss) and analyzed. At least five samples per group were analyzed by ImagePro Plus software in each experiment.

**Flow cytometry analysis**. SVFs were isolated from mice epiWAT, resuspended in PBS supplemented with 1% FBS and stained with indicated fluorescent isotope-conjugated antibodies for 30 min at room temperature in the dark. The antibodies used for FACS included anti-F4/80-BV421 (BD Biosciences, 565411) and anti-CD11b-FITC (BD Biosciences, 553310). For PMs recruitment analysis, mice were euthanized 3 days after injection of 4% sterile thioglycollate media (Sigma-Aldrich). Cells were harvested from the peritoneal cavity, washed with PBS, and incubated with PE-conjugated anti-F4/80 (R&D Systems, FAB5580P) antibody. The cells marked with the antibodies were then washed three times with PBS. Samples were analyzed using FACS Verse (BD Biosciences). For sorting F4/80$^+$ macrophages from SVFs, FACS Aria II (BD Biosciences) was used.

**Subcellular fractionation by ultracentrifugation**. Cell fractionation was done with modifications as reported[48]. After lysis of the cells in lysis buffer for 10 min on ice, cells were centrifuged at 20,000$g$ for 15 min. The post-nuclear supernatant fraction was centrifuged at 100,000$g$ for 1 h. The resulting supernatant was designated as the S fraction (supernatants, S fraction). All pellets were resuspended by lysis buffer and protease inhibitors in the original volume (pellets, P fraction). Equal volume amounts of fractions were analyzed by western bolt.

**Co-immunoprecipitation**. Cells were lysed in co-IP buffer containing protease inhibitor cocktail tablets (Roche, Germany). The cell lysates were incubated with the indicated antibody at 4 °C overnight. Next day, the cell lysates were conjugated with protein A/G beads (Santa Cruz) for 4–6 h. Immunoprecipitates were collected, washed three times in lysis buffer at 4 °C, and eluted into Laemmli sample buffer by boiling. The immunocomplex was subjected to western blot using the indicated antibodies. Antibodies applied for co-IP included anti-p65 (Cell Signaling, 8242), anti-TRAF6 (Thermo Fisher Scientific, 38-0900), anti-Flag (Sigma-Aldrich, F1804), and anti-HA (Thermo Fisher Scientific, 26183).

**RT-qPCR analysis**. Total RNA was isolated from tissues and purified cells using RNAiso Plus (TaKaRa, Japan). The quality of the RNA samples was reverse transcribed into cDNA using commercial kits (Vazyme Biotech, China). RT-qPCR was performed using the ABI Prism 7000 PCR system (Applied Biosystems) and analysis was normalized to glyceraldehyde-3-phosphate dehydrogenase (GAPDH). Primers are listed in Supplementary Table 1.

**Western blot analysis**. Cell lysates or immunoprecipitates were separated by SDS-PAGE and transferred to PVDF membranes. After blocking with 5% BSA, the membrane was incubated with primary antibodies at 4 °C overnight, followed by

incubation with the corresponding secondary antibodies for 1 h at room temperature. The membranes were washed three times for 10 min each, incubated with SuperSignal chemiluminescent substrate (Pierce) and imaged by ChemiDoc XRS$^+$ Imaging System (Bio-Rad). Blots were semi-quantified using ImageJ software. The following antibodies were used: MVP (Santa Cruz, sc-23916, 1:100), MVP (Santa Cruz, sc-18701, 1:100), phospho-AKT (Cell Signaling, 9271, 1:1000), AKT (Cell Signaling, 9272, 1:1000), phospho-IKKα/β (Cell Signaling, 2697, 1:1000), IKKα/β (Santa Cruz, sc-7607, 1:500), IκBα (Cell Signaling, 4814, 1:1000), phospho-p65 (Cell Signaling, 3033, 1:1000), p65 (Cell Signaling, 8242, 1:1000), TRAF6 (BioLegend, 654502, 1:1000), Flag (Sigma-Aldrich, F1804, 1:1000), HA (Roche, 11867423001, 1:1000), HA (Thermo Fisher Scientific, 26183, 1:5000), TRAF2 (Cell Signaling, 4724, 1:1000), TRAF3 (Santa Cruz, sc-6933, 1:200), Ub (Millipore, MAB1510, 1:1000), IRAK1 (Santa Cruz, sc-5288, 1:200), Myc (Cell Signaling, 2278, 1:1000), His (Cell Signaling, 12698, 1:1000), α-Tubulin (Protein Tech, 11224-1-AP, 1:1000), GAPDH (Kangchen Tech, KC-5G4, 1:3000), β-actin (Santa Cruz, sc-47778, 1:1000), Lamin B1 (Protein Tech, 66095-1-Ig, 1:1000). Nuclear extract preparation was conducted according to the manufacturer's instructions of a commercial kit (Thermo Fisher Scientific). Uncropped blots are available in Supplementary Fig. 12.

**Statistical analysis**. Differences between groups were examined for statistical significance using the Student's $t$ test or analysis of variance (ANOVA). All statistical tests were performed using GraphPad Prism 6.0, and all data are represented as mean ± SEM. Group comparisons were assessed with Student's $t$ test to compare two groups, and ANOVA followed by post hoc test for multiple comparisons as appropriate. For all tests, $P < 0.05$ was defined as significant.

**Reporting summary**. Further information on experimental design is available in the Nature Research Reporting Summary linked to this article.

## Data availability
All data supporting the findings of this study are available within the main manuscript and the supplementary files, or from the corresponding author upon reasonable request. A reporting summary for this article is available as a supplementary file.

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

## Acknowledgements

This work was supported by grants from the National Natural Science Foundation of China (Nos. 81830011, 81670418, and 91739304 to Q. Chen; Nos. 81870371 and 81370005 to J. Ben; No. 81770417 to X. Zhu; No. 81670263 to X. Li; No. 81500305 to H. Zhang), Natural Science Foundation of the Jiangsu Higher Education Institutions of China (18KJA310003 to J. Ben, 17KJB310004 to Yan Zhang, 15KJA310001 to X. Li), Natural Science Foundation of Jiangsu, China (BK20150048 to J. Ma); Qing Lan Project and the Project Funded by Jiangsu Province Collaborative Innovation Center for Cardiovascular Disease Translational Medicine.

## Author contributions

J. Ben and Q. Chen conceived and designed the work. B. Jiang, D. Wang, Q. Liu, Yongjing Zhang, Y. Qi, X. Tong, L. Chen, and Yan Zhang performed research, collected and analyzed the data. X. Liu collected human tissue samples. J. Ben, X. Zhu, X. Li, H. Zhang, H. Bai, Q. Yang, and J. Ma provided technical assistance. E.A.C. Wiemer provided MVP knockout mice and advice. J. Ben, B. Jiang, Q. Liu, E.A.C. Wiemer, Y. Xu, and Q. Chen wrote the paper. All authors read and approved the final manuscript.

## Additional information

**Competing interests:** The authors declare no competing interests.

