## [Peer Review File · Nature Communications]

Reviewers' comments:

Reviewer #1 (Remarks to the Author):

Review of NCOMMS-18-24966-T

“Major vault protein suppresses obesity and atherosclerosis through inhibiting IKK-NF- κ B signaling mediated inflammation” Ben et al.

This manuscript describes a new role for major vault protein (MVP) in suppressing NF- κ B signaling in macrophages. The mechanism of this action is proposed to be by inhibiting the activity of TRAF6 thereby suppressing macrophage inflammation.

Much of the data in this manuscript is quite compelling. However, I have a number of serious general concerns and some specific issues which I will address below. I want to first note that I am not a metabolism expert, and thus I was not able to fully evaluate the significance of some of the findings presented in relation to obesity. However, I have significant expertise in the vault field so I have naturally focused on this critical aspect of the manuscript.

General Concerns:

1. The authors appear to be basing their arguments on the assumption that they are studying the activity of an individual protein, MVP. However, this is a faulty assumption, as numerous studies have shown that MVP does not exist as a free protein in the cell cytoplasm, rather all cellular MVP is assembled into macromolecular vault particles that each consist of 78 MVP subunits as well as numerous copies of the vault associated proteins, TEP1 and VPARP, and the vRNA. Vaults don't self-assemble from multiple free MVP subunits: instead each vault is assembled co-translationally on polysomes resulting in assembled particles without free MVP protein intermediates. Even in vitro studies with reticulocyte lysates have shown that MVP assembles into vault particles, this is also true when full-length and tagged MVP cDNAs are expressed following transfection studies. The particulate nature of vaults is easily demonstrated by simple fractionation of cells, as vaults pellet at 100,000xg whereas free MVP proteins do not and furthermore all cellular MVP (except MVP resulting from denaturation or proteolysis during extract preparation) is found in the 100,000xg pellet.

2. The experiments show that feeding wild-type mice a high-fat diet (HFD) led to increased levels of vaults in macrophages in the stromal vascular fraction of the epididymal white adipose tissue (SVF), a similar trend was found in obese humans. Male MVP KO mice fed a HFD gained more weight, more fat tissue, increased liver size and fattier, increased levels of glucose, insulin, and other metabolic indicators. What about female mice did they exhibit the same trend?

3. Two upstream effectors of NF- κ B are IKK and p65, both are phosphorylated in LPS stimulated PMs from MVP KO mice. Nuclear extracts were analyzed and p65 levels appeared to be increased, however in the co-immunofluorescence staining, the authors suggest that the data shows co-localization of p-p65 and nuclei. However the staining of p-p65 looks to me to be more peri-nuclear and not co-localized. I believe better resolution is required to make this assertion. Perhaps confocal microscopy would be superior in this regard. In addition, no direct interaction of p65 was found with vaults by co-IP.

4. Another key regulator of NF- κ B is TRAF6. By transfecting HEK293T cells with Flag-MVP and HA-TRAF6 the authors were able to show they interact by co-IP. It is important to note here that co-IP experiments with vaults are notoriously difficult to carry out due to the particulate nature of vaults. It would have been nice if the authors had examined the expression of MVP in these cells to verify the state of the expressed protein (free or assembled), they could easily analyze the protein lysates by subcellular fractionation. The authors could show whether it was assembled vaults that interact with TRAF6 by centrifuging the cytoplasmic fraction at 100,000xg followed by analysis of the pellet (P100) by western blot. By doing this they would have been able to determine the amount of TRAF6 that co-pelleted with vaults. This type of biochemical analysis could easily be carried out as well on LPS treated cells.

5. In Fig.6K, the immunofluorescence analysis does not show good co-localization between TRAF6 and vaults, confocal microscopy should be used to confirm.

6. Line 261. "TRAF6 polyubiquitination depends on its recruitment to IRAK1 and subsequently oligo-merization." The authors claim that LPS enhanced polyubiquitination of TRAF6 in MVP KO BMDMs in Fig. 7A, however the figure is very dark and it is not very convincing.

7. The authors next set up an in vitro system where they transfected Flag MVP into HEK293T cells stating that overexpression of MVP inhibited polyubiquitination (Lines 258, 259). Did the authors investigate whether or not the HEK293T cells contain endogenous MVP? Why do the authors state that MVP is overexpressed? This would only be relevant if there is endogenous MVP.

8. In Fig. 7F-G the authors investigate the effects of truncated MVP proteins to determine which MVP domain is responsible for the TRAF6 interaction. I suspect that the truncated MVPs are either individual truncated proteins or cell aggregates that resolve when analyzed on an SDS gel. The ability of these truncated proteins or protein aggregates to interact with TRAF6 may be entirely different from vault particle interaction with TRAF6.

9. These final figures are crucial to the authors' arguments yet the description of the experiments are so brief that they are difficult to follow.

Specific comments:

Line 117 The authors should comment on the reduced levels of leptin and adiponectin mRNA levels in MVP KO mice.

Line 112. What is the relationship of suppressing AKT phosphorylation to impairing glucose and insulin tolerance?

Lines 208-209. Please explain what the significance is of Ly-6Chi monocyte labeling.

Line 211. How does thioglycolate work and how is it administered? There is no mention of it in the Methods.

Lines 223-224. Increased production of TNF- α and CCL2 in LPS treated MVP KO macrophages parallels the elevated plasma levels observed in the MacMVP-ApoE- mice, therefore these studies suggest that MVP deficiency results in a pro-inflammatory environment. The final sentence should be moved to follow after (Fig. 5S) in line 220.

Line 233. Refers to nuclear translocation of p65 (Fig. 6D-E), there is no description of nuclear extract preparation in the Methods and the merged images do not look nuclear, the p-p65 images look perinuclear. The nuclei that are stained with DAPI are distinct spots, whereas the p-p65 images are much more diffuse in comparison. This should be analyzed by confocal microscopy for better resolution.

Line 649. In Fig.6 PMs treated with LPS from KO mice again show elevated levels of both mRNA and protein levels TNF- α and CCL2. Were these normal weight mice or obese mice (the legend does not indicate)?

Line 650. Figure 6 A. No indicated times as stated in the legend. Are these PM's from normal weight or obese mice?

Reviewer #2 (Remarks to the Author):

Key results

The authors present data suggesting a role for major vault protein as a suppressor of NF-kB activation in macrophages during HF-feeding. Data from MVP KO mice and myeloid specific MVP deletion is associated with increased weight gain, insulin resistance and hepatic steatosis. Furthermore, the authors propose a mechanism by which MVP attenuates NF-kB activation which involves the attenuation of TRAF6 dependent recruitment of IRAK-1.

Validity

The in vivo studies which utilize genetic mouse models (MVP KO mice and myeloid-specific MVP deletion) are solid. They utilize standard and well-validated HF-feeding models and present the standard endpoints that many investigators have utilized. The atherosclerosis model (apoE^{-/-}) is also a standard model.

The senior author has championed the role of MVP in regulating inflammatory responses during chronic low-grade inflammation. He has published (JBC 2013) utilizing these genetic mouse models.

Overall, the models and data presented are solid.

Significance

In this paper the authors have uncovered a novel role for MVP in regulating macrophage activation during obesity/atherosclerosis. The role of macrophage activation is well described, however, regulatory mechanisms underlying these responses remains an important and unanswered question. The in vivo data utilizing the HF-feeding and atherosclerosis models are compelling and support the primary hypothesis. The effect of MVP deletion in two separate well established disease models is also compelling.

These studies/results will be of interest to the obesity community.

Data and methods

There is a tremendous amount of data presented in figures 1-5. These data are solid and compelling. I do not have additional technical comments or suggestions.

The main weakness of the manuscript is the last figure 6. The authors are trying to establish a mechanism by which MVP modulates NF-kB activation. Figure 6 (A-E) establishes the standard in vitro model (LPS-stimulation) utilizing PM from WT and MVP KO mice. An interaction is suggested by co-IP experiments with TRAF6. These data are suggestive of possible mechanism however much more experiments are needed to establish this as the “mechanism” I

Nevertheless, this reviewer feels that the in vivo data from figures 1-5 are sufficient to establish a role for macrophage MVP. The preliminary mechanistic studies should be saved for a separate manuscript and that the current paper should emphasize the first 5 figures.

Additional Experiments

None

Reviewer #3 (Remarks to the Author):

Ben and colleagues found a significant upregulation of MVP in macrophages from obese mice and overweight/obese human subjects. By examining global and conditional myeloid-specific KO mice, they provide evidence that MVP suppressed many effects associated with high fat diet (HFD) such as obesity, insulin resistance, liver steatosis and atherosclerosis. Specifically, MVP in macrophages antagonized inflammatory responses and NF- κ B signaling. Association and functional studies suggest that MVP acted as a negative regulator of the E3 ligase TRAF6, which is a key regulator of NF- κ B in response to different innate and pro-inflammatory stimuli. The authors present very clear proof for the aggravation of obesity associated diseases in the two MVP KO mouse models. These data define MVP as a new player controlling inflammation in metabolic diseases.

Major comments:

If MVP acts as a mere suppressor of inflammatory signaling, it is at least puzzling that MVP expression is induced in murine and human macrophages after the onset of obesity. Why does upregulation of the negative regulator MVP fail to suppress inflammation? Do the data imply that MVP under HFD is unable to fulfill its suppressor function? The authors should test if NF- κ B signaling and cytokine expression in isolated macrophages CD and HFD mice is different. Further, would failure to upregulate MVP worsen obesity and inflammation? There seems to be a considerable heterogeneity in the induction of MVP in human obese subjects. Is it possible to show any correlation that less MVP correlates with enhanced inflammation markers? The discrepancy between the substantial expression and the negative regulatory function of MVP needs to be addressed and discussed with regard to the general concept.

It seems clear that basal and LPS-induced NF- κ B signaling and cytokine expression is augmented in MVP KO macrophages. However, a specific involvement of TRAF6 has not been rigorously proven. It may be beyond the scope of the manuscript to provide genetic proof that TRAF6 is the critical downstream factor, but some experiments should be performed to confirm the critical requirement of MVP for TRAF6-dependent NF- κ B activation (see specific comments to Figure 6 and 7).

Specific comments:

Figure 1: Where male or female mice and human subjects included for analyses of MVP expression? Is there any gender bias? For human individuals the normal weight group was compared to overweight/obese. How was this defined?

Figure 6: A and B) The authors should test TNF α stimulation, which is independent of TRAF6 to see whether MVP KO is selectively enhancing TRAF6-dependent LPS signaling. D) MVP has been associated with nucleocytoplasmic transport. Does enhanced p65 translocation correlate with a decrease in cytosolic I κ B α levels as a result from augmented IKK activation? G) The signal of TRAF6-bound MVP migrates slightly slower compared to MVP in the lysates. The TRAF6 IP should be repeated and MVP KO PMs should be included to confirm that the band is MVP. H) Co-IPs between Flag MVP and HA TRAF6 should include the proper transfection control HA TRAF6 with Flag empty vector for the Flag IP to confirm specificity. Further, is MVP selectively binding to TRAF6 or also associating with other TRAFs, e.g. TRAF2 and TRAF3? J) Enhanced binding of TRAF6 and MVP after LPS seems rather weak (I). How was this quantified and how as normalization performed? It would be interesting to compare TRAF6-MVP binding under conditions of induced MVP expression comparing macrophages from CD and HFD treated mice (see major comment 1).

Figure 7: A) The Ub Western Blot is overexposed and basically shows augmented basal TRAF6 ubiquitination already without stimulation. Same is true for IRAK1 binding in C. The somewhat unselective binding of MVP and TRAF6 is at least puzzling (Supp. Figure 6). To validate the conclusions that the destruction of TRAF6 oligomerization and ubiquitination by MVP 686-870 is critical, the authors need to test LPS-triggered induction of gene expression also after overexpression of MVP 1-480 and MVP 481-685. Despite the fact that these fragments bind to TRAF6, they should not affect gene expression after LPS. To confirm TRAF6 dependency, it needs to be tested if induction of cytokines after TNF α treatment is not antagonized by MVP and MVP 686-870.

Reviewers' comments:

Reviewer #1 (Remarks to the Author):

Review of NCOMMS-18-24966-T“Major vault protein suppresses obesity and atherosclerosis through inhibiting IKK-NF- κ B signaling mediated inflammation” Ben et al.

This manuscript describes a new role for major vault protein (MVP) in suppressing NF- κ B signaling in macrophages. The mechanism of this action is proposed to be by inhibiting the activity of TRAF6 thereby suppressing macrophage inflammation.

Much of the data in this manuscript is quite compelling. However, I have a number of serious general concerns and some specific issues which I will address below. I want to first note that I am not a metabolism expert, and thus I was not able to fully evaluate the significance of some of the findings presented in relation to obesity. However, I have significant expertise in the vault field so I have naturally focused on this critical aspect of the manuscript.

General Concerns:

1. The authors appear to be basing their arguments on the assumption that they are studying the activity of an individual protein, MVP. However, this is a faulty assumption, as numerous studies have shown that MVP does not exist as a free protein in the cell cytoplasm, rather all cellular MVP is assembled into macromolecular vault particles that each consist of 78 MVP subunits as well as numerous copies of the vault associated proteins, TEP1 and VPARP, and the vRNA. Vaults don't self-assemble from multiple free MVP subunits: instead each vault is

assembled co-translationally on polysomes resulting in assembled particles without free MVP protein intermediates. Even in vitro studies with reticulocyte lysates have shown that MVP assembles into vault particles, this is also true when full-length and tagged MVP cDNAs are expressed following transfection studies. The particulate nature of vaults is easily demonstrated by simple fractionation of cells, as vaults pellet at 100,000xg whereas free MVP proteins do not and furthermore all cellular MVP (except MVP resulting from denaturation or proteolysis during extract preparation) is found in the 100,000xg pellet.

A: Thanks for the reviewer's professional and valuable suggestions. We accepted the reviewer's indication and newly performed the cell fractionation experiments. New data have been added to the revised manuscript. The Introduction, Results and Discussion sections (line 63, line 254-267, line 319, line 350) have been re-edited.

2. The experiments show that feeding wild-type mice a high-fat diet (HFD) led to increased levels of vaults in macrophages in the stromal vascular fraction of the epididymal white adipose tissue (SVF), a similar trend was found in obese humans. Male MVP KO mice fed a HFD gained more weight, more fat tissue, increased liver size and fattier, increased levels of glucose, insulin, and other metabolic indicators. What about female mice did they exhibit the same trend?

A: We newly determined the expressional level of MVP in macrophages of gonadal WAT (gonWAT) SVFs from the HFD-fed female mice. MVP was up-regulated significantly compare with CD-fed female mice (line 94, Supplementary Fig. 1A).

We also examined the phenotype of female MVP KO mice fed a HFD (Supplementary Fig. 3, line 135). The results reveal that HFD-fed female MVP KO mice exhibited similar phenotypic changes to those male mice.

3. Two upstream effectors of NF- κ B are IKK and p65, both are phosphorylated in LPS stimulated PMs from MVP KO mice. Nuclear extracts were analyzed and p65 levels appeared to be increased, however in the co-immunofluorescence staining, the authors suggest that the data shows co-localization of p-p65 and nuclei. However the staining of p-p65 looks to me to be more peri-nuclear and not co-localized. I believe better resolution is required to make this assertion. Perhaps confocal microscopy would be superior in this regard. In addition, no direct interaction of p65 was found with vaults by co-IP.

A: Per the reviewer's suggestion, we re-conducted the IF experiment with p65 antibody and the images were observed under confocal microscope. The new data showed clearly co-localization of p65 and nuclei (Fig 6G).

4. Another key regulator of NF- κ B is TRAF6. By transfecting HEK293T cells with Flag-MVP and HA-TRAF6 the authors were able to show they interact by co-IP. It is important to note here that co-IP experiments with vaults are notoriously difficult to carry out due to the particulate nature of vaults. It would have been nice if the authors had examined the expression of MVP in these cells to verify the state of the expressed protein (free or assembled), they could easily analyze the protein lysates by

subcellular fractionation. The authors could show whether it was assembled vaults that interact with TRAF6 by centrifuging the cytoplasmic fraction at 100,000xg followed by analysis of the pellet (P100) by western blot. By doing this they would have been able to determine the amount of TRAF6 that co-pelleted with vaults. This type of biochemical analysis could easily be carried out as well on LPS treated cells.

A: We did experiments as the reviewer indicated. Flag-MVP and HA-TRAF6 were co-transfected into HEK293T cells. After harvest and ultracentrifugation, Flag-MVP was mainly detected in the pellets but not in the supernatants, suggesting that Flag-MVP should exist as assembled vaults. Meanwhile, HA-TRAF6 could be co-precipitated with the assembled MVP in the pellets (line 259, Supplementary Fig.9D). Similarly, TRAF6 could be co-precipitated with assembled MVP in macrophages by ultracentrifugation, which was enhanced by LPS stimulation (line 254, line 264, Supplementary Fig.9B, E).

5. In Fig.6K, the immunofluorescence analysis does not show good co-localization between TRAF6 and vaults, confocal microscopy should be used to confirm.

A: We newly did the experiment by using the confocal microscopy. New figures are presented in Fig 6M.

6. Line 261. "TRAF6 polyubiquitination depends on its recruitment to IRAK1 and subsequently oligo-merization." The authors claim that LPS enhanced polyubiquitination of TRAF6 in MVP KO BMDMs in Fig. 7A, however the figure is

very dark and it is not very convincing.

A: Yes, we re-conducted the experiment and the better figures are presented in Fig 7A.

7. The authors next set up an in vitro system where they transfected Flag MVP into HEK293T cells stating that overexpression of MVP inhibited polyubiquitination (Lines 258, 259). Did the authors investigate whether or not the HEK293T cells contain endogenous MVP? Why do the authors state that MVP is overexpressed? This would only be relevant if there is endogenous MVP.

A: We examined the expression of MVP in HEK293T cells by western blot. The results showed that HEK293T cells do express endogenous MVP but expression level is lower than that in RAW264.7 macrophages. The western blots are presented as follows.

8. In Fig. 7F-G the authors investigate the effects of truncated MVP proteins to determine which MVP domain is responsible for the TRAF6 interaction. I suspect that the truncated MVPs are either individual truncated proteins or cell aggregates that resolve when analyzed on an SDS gel. The ability of these truncated proteins or

protein aggregates to interact with TRAF6 may be entirely different from vault particle interaction with TRAF6.

A: The full length and truncated MVPs were transfected into HEK293T cells, harvested and centrifuged at 100,000 g. The western blot showed that the full length MVP was primarily in the pellets, while the truncated MVPs were detected in both supernatants and pellets (line 295, Supplementary Fig. 10E). At the moment we still don't know what forms of MVP truncates work in cells. However, all three MVP truncates could bind to TRAF6, but only MVP-FL and MVP α -helical domain (686-870) could substantially inhibit the TRAF6 oligomerization, self-ubiquitination, the LPS-induced production of TNF- α and CCL2, and the nuclear translocation of p65 in macrophages. These results suggest that both MVP truncates and full length MVP function in cells.

9. These final figures are crucial to the authors' arguments yet the description of the experiments are so brief that they are difficult to follow.

A: Yes, we re-edited the Results and Legends section of the final figures as the reviewer suggested.

Specific comments:

Line 117 The authors should comment on the reduced levels of leptin and adiponectin mRNA levels in MVP KO mice.

A: Leptin and adiponectin are the most abundant cytokines secreted from adipocytes

(adipokines). They generally regulate energy homeostasis, increase insulin sensitivity and suppress inflammation, possessing pleiotropic salutary effects against a cluster of obesity-related disorders (*J Clin Invest.* 2003,111:1409-21; *J Clin Invest.* 2006,116:1784-92). Their levels normally reflect the function of adipocytes. The reduced levels of leptin and adiponectin in MVP KO mice may reflect adipocyte dysfunction. We annotated it in the results (line 116).

Line 112. What is the relationship of suppressing AKT phosphorylation to impairing glucose and insulin tolerance?

A: The metabolic effects of insulin are largely dependent on PI3K-AKT signaling in target tissues including adipose tissue, liver and skeletal muscle (*Cell Metab.* 2011, 14:575-85). The phosphorylation of AKT, a readout of intracellular insulin signaling, in target tissues was inhibited when the glucose and insulin tolerance was impaired. We annotated it in the results (line 123).

Lines 208-209. Please explain what the significance is of Ly-6Chi monocyte labeling.

A: Ly-6C high expression (Ly-6C^{hi}) is a pro-inflammatory monocyte marker (*Cir Res.*2018,122:113-27). We used “Ly-6C^{hi} pro-inflammatory monocytes” instead of “Ly-6C^{hi} subset of monocytes” in the revised manuscript (line 211).

Line 211. How does thioglycolate work and how is it administered? There is no mention of it in the Methods.

A: Peritoneal injection of thioglycolate media could induce inflammation and mono-macrophage recruitment in mice (*J Clin Invest.* 2007,117: 185-94; *J Immunol Methods.* 1996, 197:139-50). We injected intra-peritoneally 1 ml of sterile 4% thioglycolate media and harvested the cells from the murine peritoneal cavities 3 days later. The description was added into the Results (line 214), Figure legends (line 668) and Methods of supplementary data.

Lines 223-224. Increased production of TNF- α and CCL2 in LPS treated MVP KO macrophages parallels the elevated plasma levels observed in the MacMVP-ApoE-mice, therefore these studies suggest that MVP deficiency results in a pro-inflammatory environment. The final sentence should be moved to follow after (Fig. 5S) in line 220.

A: Yes, we re-edited the manuscript per the reviewer's suggestion (line 224).

Line 233. Refers to nuclear translocation of p65 (Fig. 6D-E), there is no description of nuclear extract preparation in the Methods and the merged images do not look nuclear, the p-p65 images look perinuclear. The nuclei that are stained with DAPI are distinct spots, whereas the p-p65 images are much more diffuse in comparison. This should be analyzed by confocal microscopy for better resolution.

A: The cell nuclear extract preparation was conducted according to the manufacturer's instructions of a commercial kit (Thermo Fisher Scientific, Prod No. 78833, NE-PRE Nuclear and Cytoplasmic Extraction Reagents). This information

was added into the Methods of supplemental data.

Also, we have re-conducted IF experiment as the reviewer indicated in the 3rd point of general concerns and better images are presented in Fig 6G.

Line 649. In Fig.6 PMs treated with LPS from KO mice again show elevated levels of both mRNA and protein levels TNF- α and CCL2. Were these normal weight mice or obese mice (the legend does not indicate)?

A: In Fig. 6 PMs were isolated from normal weight mice fed by a CD. We re-edited the figure legends to make it clearer (line 680).

Line 650. Figure 6 A. No indicated times as stated in the legend. Are these PM's from normal weight or obese mice?

A: The PMs were treated by LPS for 12 h. They were collected from normal weight mice fed by a CD. The information was given in the figure legends (line 680-681).

Reviewer #2 (Remarks to the Author):

Key results

The authors present data suggesting a role for major vault protein as a suppressor of NF- κ B activation in macrophages during HF-feeding. Data from MVP KO mice and myeloid specific MVP deletion is associated with increased weight gain, insulin resistance and hepatic steatosis. Furthermore, the authors propose a mechanism by which MVP attenuates NF- κ B activation which involves the attenuation of TRAF6

dependent recruitment of IRAK-1.

Validity

The in vivo studies which utilize genetic mouse models (MVP KO mice and myeloid-specific MVP deletion) are solid. They utilize standard and well-validated HF-feeding models and present the standard endpoints that many investigators have utilized. The atherosclerosis model (apoE^{-/-}) is also a standard model.

The senior author has championed the role of MVP in regulating inflammatory responses during chronic low-grade inflammation. He has published (JBC 2013) utilizing these genetic mouse models. Overall, the models and data presented are solid.

Significance

In this paper the authors have uncovered a novel role for MVP in regulating macrophage activation during obesity/atherosclerosis. The role of macrophage activation is well described, however, regulatory mechanisms underlying these responses remains an important and unanswered question. The in vivo data utilizing the HF-feeding and atherosclerosis models are compelling and support the primary hypothesis. The effect of MVP deletion in two separate well established disease models is also compelling. These studies/results will be of interest to the obesity community.

Data and methods

There is a tremendous amount of data presented in figures 1-5. These data are solid and compelling. I do not have additional technical comments or suggestions.

The main weakness of the manuscript is the last figure 6. The authors are trying to

establish a mechanism by which MVP modulates NF- κ B activation. Figure 6 (A-E) establishes the standard in vitro model (LPS-stimulation) utilizing PM from WT and MVP KO mice. An interaction is suggested by co-IP experiments with TRAF6. These data are suggestive of possible mechanism however much more experiments are needed to establish this is as the “mechanism” I

Nevertheless, this reviewer feels that the in vivo data from figures 1-5 are sufficient to establish a role for macrophage MVP. The preliminary mechanistic studies should be saved for a separate manuscript and that the current paper should emphasize the first 5 figures.

Additional Experiments

None

A: Thanks a lot for the reviewer’s appreciations. We have newly conducted more experiments including cell fractionation and treatment of macrophages by TNF α . New results have been added to Fig 6-7 and supplementary figures. The text has been re-edited.

Reviewer #3 (Remarks to the Author):

Ben and colleagues found a significant upregulation of MVP in macrophages from obese mice and overweight/obese human subjects. By examining global and conditional myeloid-specific KO mice, they provide evidence that MVP suppressed many effects associated with high fat diet (HFD) such as obesity, insulin resistance, liver steatosis and atherosclerosis. Specifically, MVP in macrophages antagonized

inflammatory responses and NF- κ B signaling. Association and functional studies suggest that MVP acted as a negative regulator of the E3 ligase TRAF6, which is a key regulator of NF- κ B in response to different innate and pro-inflammatory stimuli. The authors present very clear proof for the aggravation of obesity associated diseases in the two MVP KO mouse models. These data define MVP as a new player controlling inflammation in metabolic diseases.

Major comments:

If MVP acts as a mere suppressor of inflammatory signaling, it is at least puzzling that MVP expression is induced in murine and human macrophages after the onset of obesity. Why does upregulation of the negative regulator MVP fail to suppress inflammation? Do the data imply that MVP under HFD is unable to fulfill its suppressor function? The authors should test if NF- κ B signaling and cytokine expression in isolated macrophages CD and HFD mice is different.

Further, would failure to upregulate MVP worsen obesity and inflammation? There seems to be a considerable heterogeneity in the induction of MVP in human obese subjects. Is it possible to show any correlation that less MVP correlates with enhanced inflammation markers? The discrepancy between the substantial expression and the negative regulatory function of MVP needs to be addressed and discussed with regard to the general concept.

A: It is intrigued that MVP expression is induced in murine and human macrophages after the onset of obesity. Per the reviewer's suggestion, we newly measured the expression of MVP and NF- κ B signaling in the isolated murine epiWAT macrophages

of CD and HFD mice. We found that the activated degree of NF- κ B signaling and over-production of inflammatory cytokines were much stronger in HFD mice than that in CD mice. However, MVP up-regulation was mild (line 241, Supplementary Fig. 8B, C). This may reflect that HFD-induced insufficient MVP expression would be unable to completely fulfill its suppressing inflammation function in macrophages.

Further, we collected more overweight/obese human visceral adipose tissues and measured levels of MVP, TNF α , and CCL2 in CD14⁺ macrophages. The results showed that the expression of MVP was negatively correlated with CCL2 (line 244, Fig. 6I), revealing the general relationship between the expression of MVP and inflammatory response in the body. We thought that the observed up-regulation of MVP in obesity-associated metabolic disorders and atherosclerotic lesions may be elicited by inflammation, because the pro-inflammatory transcription factors such as SP1 and STAT1 can bind to the promoter of MVP and, thus, to promote its expression. On the other side, our results showed that MVP seems not to interact with TRAF2 or TRAF3 (line 254, Supplementary Fig. 9C) and may not influence the TNF α -induced pro-inflammatory cytokines production in macrophages (line 227, Supplementary Fig. 7A). The selective inhibition of NF- κ B up-stream signaling reveals that MVP may be unable to suppress metabolic inflammation completely. This may partly explain the result that the MVP expression was negatively correlated with CCL2 but not with TNF- α in obese human macrophages. Thus, MVP may constitute an essential constraint in a negative feedback loop to fine-tune inflammatory responses in macrophages, that may contribute to “low grade and chronic” metabolic inflammation.

We added the new data to the Results section and discussed them (line 362-370).

It seems clear that basal and LPS-induced NF- κ B signaling and cytokine expression is augmented in MVP KO macrophages. However, a specific involvement of TRAF6 has not been rigorously proven. It may be beyond the scope of the manuscript to provide genetic proof that TRAF6 is the critical downstream factor, but some experiments should be performed to confirm the critical requirement of MVP for TRAF6-dependent NF- κ B activation (see specific comments to Figure 6 and 7).

A: Yes, we conducted more experiments to confirm the role of MVP in TRAF-6-dependent NF- κ B signaling. New results have been presented in Supplementary Fig. 7A, Supplementary Fig. 9C and Supplementary Fig. 11A.

Specific comments:

Figure 1: *Where male or female mice and human subjects included for analyses of MVP expression? Is there any gender bias?*

A: We newly examined the expression of MVP in gonWAT macrophages and PMs from female mice. It was up-regulated in the HFD-fed female mice compare to the CD-fed female mice, which is similar to that in male mice (line 93, Supplementary Fig. 1A, B).

We also collected more human visceral adipose tissues to examine the expressional level of MVP in the isolated macrophages. MVP was up-regulated in both male and female overweight/obese subjects compared with normal weight subjects, but there

was no statistical significance by ANOVA analysis due to the limited number of human subjects (the figure below).

These data suggest that obesity should induce up-regulation of MVP in both male and female ones. We have added it to the Result section (line 95).

For human individuals the normal weight group was compared to overweight/obese.

How was this defined?

A: The groups of human individuals were defined by BMI as follows: normal weight ($18.5 \leq \text{BMI} < 24$), overweight ($24 \leq \text{BMI} < 28$) or obese ($\text{BMI} \geq 28$). This information was added into the legend of Figure 1J (line 570).

Figure 6: A and B) The authors should test TNF α stimulation, which is independent of TRAF6 to see whether MVP KO is selectively enhancing TRAF6-dependent LPS signaling.

A: We did the experiment per the reviewer's indication. TNF α and CCL2 were up-regulated in macrophages treated with TNF α . MVP deficiency did not alter the TNF- α -induced pro-inflammatory cytokines production in macrophages. These

results were added into the Results section (line 227, Supplementary Fig. 7A) and discussed in line 373.

D) MVP has been associated with nucleocytoplasmic transport. Does enhanced p65 translocation correlate with a decrease in cytosolic IκBα levels as a result from augmented IKK activation?

A: Per the review's suggestion, we newly determined the cytosolic IκBα level in macrophages by western blot. It was shown that the degradation of IκBα was induced by LPS treatment and enhanced in MVP KO macrophages (line 235, Fig 6C, D). Our results demonstrated an association between a decrease in cytosolic IκBα levels and augmented IKK and p65 activation in macrophages.

G) The signal of TRAF6-bound MVP migrates slightly slower compared to MVP in the lysates. The TRAF6 IP should be repeated and MVP KO PMs should be included to confirm that the band is MVP.

A: We repeated the experiment as the reviewer suggested. New results were presented in Fig 6J.

H) Co-IPs between Flag MVP and HA TRAF6 should include the proper transfection control HA TRAF6 with Flag empty vector for the Flag IP to confirm specificity. Further, is MVP selectively binding to TRAF6 or also associating with other TRAFs, e.g. TRAF2 and TRAF3?

A: We added the transfection control HA-TRAF6 with Flag empty vector for the Flag IP, and Flag-MVP with HA empty vector for the HA IP. These results were added in Fig 6K.

Per the suggestion by reviewer 1, we also performed the cell fractionation experiments. We found that only TRAF6 but not TRAF2 or TRAF3 could be co-precipitated with MVP in the cell pellet by ultracentrifugation at 100,000 g (line 254, Supplementary Fig. 9B, C), revealing that MVP/vaults may be selectively binding to TRAF6 but not to TRAF2 or TRAF3.

J) Enhanced binding of TRAF6 and MVP after LPS seems rather weak (I). How was this quantified and how as normalization performed? It would be interesting to compare TRAF6-MVP binding under conditions of induced MVP expression comparing macrophages from CD and HFD treated mice (see major comment 1).

A: We re-conducted the experiments and the results were re-analyzed (Fig 6L). MVP and TRAF6 were co-precipitated by TRAF6 antibody, detected by western blot and analyzed by Image J. The amount of MVP-TRAF6 complex formation was quantified by the normalization of MVP to TRAF6.

EpiWAT macrophages were isolated by flow cytometry. It is difficult to get enough macrophages and proteins for the IP experiment. Instead, we isolated epiWAT SVFs from CD- and HFD-fed mice for this experiment. The results showed that MVP expression and TRAF6-MVP binding were all enhanced in HFD-fed mice (line 266, Supplementary Fig. 9F).

Figure 7: A) *The Ub Western Blot is overexposed and basically shows augmented basal TRAF6 ubiquitination already without stimulation. Same is true for IRAK1 binding in C.*

A: The experiments were re-conducted, and better western blots were presented in Fig 7A and C.

The somewhat unselective binding of MVP and TRAF6 is at least puzzling (Supp. Figure 6). To validate the conclusions that the destruction of TRAF6 oligomerization and ubiquitination by MVP 686-870 is critical, the authors need to test LPS-triggered induction of gene expression also after overexpression of MVP 1-480 and MVP 481-685. Despite the fact that these fragments bind to TRAF6, they should not affect gene expression after LPS. To confirm TRAF6 dependency, it needs to be tested if induction of cytokines after TNF α treatment is not antagonized by MVP and MVP 686-870.

A: We conducted all the experiments as the reviewer indicated. The LPS-triggered expression of TNF α and CCL2 in macrophages was not affected significantly by over-expression of MVP-(1-480) or MVP-(481-685) (line 304, Supplementary Fig. 11B, C).

Further, the induction of cytokines by TNF α was not antagonized obviously by MVP or MVP-(686-870) over-expression (line 302, Supplementary Fig. 11A), reflecting that MVP and MVP-(686-870) may have no impact on the

TNF- α -TRAF2/3 signaling in macrophages. These results together with the above-mentioned results have been discussed in the revised manuscript (line 373).

REVIEWERS' COMMENTS:

Reviewer #1 (Remarks to the Author):

This manuscript describes a new role for major vault protein (MVP) in suppressing NF- κ B signaling in macrophages. The mechanism of this action is proposed to be by inhibiting the activity of TRAF6 thereby suppressing macrophage inflammation.

Much of the data in this manuscript is quite compelling and the authors have addressed the concerns expressed by this reviewer previously. I believe the findings are novel and will be of interest to the scientific community.

Reviewer #3 (Remarks to the Author):

The authors have done a very good job and thoroughly addressed my critiques concerning the mechanism. They included new convincing data that strengthen the assumption that MVP affects NF- κ B signaling in a TRAF6-dependent manner. It is still puzzling that MVP as a suppressor is upregulated in HFD mice and obese subjects. Nevertheless, the authors establish a compelling new role of MVP in obesity and inflammatory signaling. I have no further comments.

REVIEWERS' COMMENTS:

Reviewer #1 (Remarks to the Author):

This manuscript describes a new role for major vault protein (MVP) in suppressing NF- κ B signaling in macrophages. The mechanism of this action is proposed to be by inhibiting the activity of TRAF6 thereby suppressing macrophage inflammation.

Much of the data in this manuscript is quite compelling and the authors have addressed the concerns expressed by this reviewer previously. I believe the findings are novel and will be of interest to the scientific community.

A: Thank the reviewer for the comments.

Reviewer #3 (Remarks to the Author):

The authors have done a very good job and thoroughly addressed my critiques concerning the mechanism. They included new convincing data that strengthen the assumption that MVP affects NF- κ B signaling in a TRAF6-dependent manner. It is still puzzling that MVP as a suppressor is upregulated in HFD mice and obese subjects. Nevertheless, the authors establish a compelling new role of MVP in obesity and inflammatory signaling. I have no further comments.

A: Thank the reviewer for the valuable suggestions. The issue of up-regulation of MVP in obesity is worthy further investigation in the future.